# Metabolic memory of Δ9-tetrahydrocannabinol exposure in pluripotent stem cells and primordial germ cells-like cells

Roxane Verdikt[1], Abigail A Armstrong[2], Jenny Cheng[3], Young Sun Hwang[4], Amander T Clark[4,5,6], Xia Yang[7,8], Patrick Allard[1,9]*

[1]Institute for Society and Genetics, University of California, Los Angeles, Los Angeles, United States; [2]Department of Obstetrics/Gynecology and Reproductive Endocrinology and Infertility, University of California, Los Angeles, Los Angeles, United States; [3]Molecular, Cellular, and Integrative Physiology Graduate Program, University of California, Los Angeles, Los Angeles, United States; [4]Department of Molecular Cell and Developmental Biology, University of California, Los Angeles, Los Angeles, United States; [5]Center for Reproductive Science, Health and Education, University of California, Los Angeles, Los Angeles, United States; [6]Eli and Edythe Broad Center of Regenerative Medicine and Stem Cell Research, University of California, Los Angeles, Los Angeles, United States; [7]Integrative Biology and Physiology Department, University of California, Los Angeles, Los Angeles, United States; [8]Department of Molecular and Medical Pharmacology, University of California, Los Angeles, Los Angeles, United States; [9]Molecular Biology Institute, University of California, Los Angeles, Los Angeles, United States

*For correspondence:
pallard@ucla.edu

Competing interest: The authors declare that no competing interests exist.

**Abstract** Cannabis, the most consumed illicit psychoactive drug in the world, is increasingly used by pregnant women. However, while cannabinoid receptors are expressed in the early embryo, the impact of phytocannabinoids exposure on early embryonic processes is lacking. Here, we leverage a stepwise in vitro differentiation system that captures the early embryonic developmental cascade to investigate the impact of exposure to the most abundant phytocannabinoid, Δ9-tetrahydrocannabinol (Δ9-THC). We demonstrate that Δ9-THC increases the proliferation of naive mouse embryonic stem cells (ESCs) but not of their primed counterpart. Surprisingly, this increased proliferation, dependent on the CB1 receptor binding, is only associated with moderate transcriptomic changes. Instead, Δ9-THC capitalizes on ESCs' metabolic bivalence by increasing their glycolytic rates and anabolic capabilities. A memory of this metabolic rewiring is retained throughout differentiation to Primordial Germ Cell-Like Cells in the absence of direct exposure and is associated with an alteration of their transcriptional profile. These results represent the first in-depth molecular characterization of the impact of Δ9-THC exposure on early stages of germline development.

## eLife assessment

The study presents **valuable** findings demonstrating that physiologically relevant concentrations delta-9-tetrahydrocannabinol, which is found in cannabis, have metabolic effects on early mouse embryonic cell types. The evidence supporting the claims is **convincing**. The work will be of interest to researchers in stem cell and epigenetics fields.

**eLife digest** Cannabis is the most widely used illicit drug in the world, with 4.3% of the global adult population estimated to have used it in the previous year. In particular, the consumption of cannabis by pregnant women has almost doubled in recent years and is particularly increased in those aged under 18.

The main psychoactive component in cannabis, known as Δ9-THC, activates cannabinoid receptors in the brain, including the receptor CB1. Recent research has shown that CB1 is also active in the mouse embryo, but it remained unclear if Δ9-THC could also affect the development of an embryo.

To better understand the potential effects of this exposure, scientists can study stem cells that develop into germ cells (which go on to form egg and sperm), which have been grown in the laboratory. Emerging research has shown that germ cells are particularly sensitive to changes in their environment and due to their role in reproduction, changes can have knock-on effects for embryos.

Verdikt et al. studied the effects of Δ9-THC on mouse embryonic stem cells, finding that it caused them to multiply more quickly. This was dependent on both Δ9-THC binding to the CB1 receptor that causes the psychoactive effects of cannabis in the brain and an increased energy metabolism. Blocking an important metabolic pathway called glycolysis caused the Δ9-THC-treated cells to return to a normal multiplication rate. The exposed stem cells also gave rise to germ cells with abnormal metabolism and altered gene expression, suggesting that this metabolic 'memory' can be passed on to cells in the next developmental stage.

Overall, the findings indicate that exposure to Δ9-THC alters the metabolism in early embryonic cells of mice and that these effects can be lasting. This emphasises the need for further research on the impact of cannabis use during pregnancy, particularly as the drug's availability is expected to increase significantly with changes in regulation. The work also contributes to research highlighting the inheritance of metabolism.

## Introduction

Cannabis is the most widely used illicit psychoactive drug in the world (***U.N Office on Drugs and Crime, 2022***). In the United States, an estimated 49.6 million people, roughly 18% of the population, consumed cannabis at least once in 2020, with indications that these numbers will likely increase in the coming years as attitudes and regulations change (***Mennis et al., 2023***; ***Substance Abuse and Mental Health Services Administration, 2020***). In particular, between 7% and 12% of expecting women report cannabis use, predominantly during the first trimester, to alleviate the symptoms of morning sickness (***Chabarria et al., 2016***; ***Volkow et al., 2019***; ***Young-Wolff et al., 2018***). These statistics indicate that a significant number of developing embryos are exposed to cannabis, with limited knowledge of the biological repercussions of such exposure.

Among the several hundred unique phytocannabinoids present in Cannabis sativa, (−)-trans-Δ9-t etrahydrocannabinol (Δ9-THC) is chiefly responsible for the psychoactivity of cannabis (***Andre et al., 2016***). As a result, the level of Δ9-THC in recreational cannabis has increased over the last 10 years and now commonly accounts for 20% of total compounds (***Chandra et al., 2019***). The psychoactive effects of Δ9-THC arise from its binding and subsequent activation of the G-protein-coupled cannabinoid receptor CB1 largely expressed in the central nervous system (***Pacher et al., 2020***). In this context, Δ9-THC exposure has been shown to durably alter metabolic, transcriptional, and epigenetic programs in the brain (***Bénard et al., 2012***; ***Prini et al., 2018***; ***Szutorisz and Hurd, 2018***; ***Watson et al., 2015***). While over the last decades, significant attention has been paid to Δ9-THC's neurological effects, there is also evidence, albeit more limited, of its impact on reproductive functions (***Lo et al., 2022***). Data shows CB1 expression in the male and the female reproductive tracts, in the pre-implantation embryo and in the placenta (***Lo et al., 2022***; ***Paria et al., 1995***). In animal models as well as in humans, exposure to cannabis is associated with reduced fertility, decreased testis weight and sperm count, and impairment of embryo implantation (***Lo et al., 2022***). In males, these effects are correlated with an alteration of the sperm transcriptome and epigenome (***Murphy et al., 2018***; ***Osborne et al., 2020***; ***Schrott and Murphy, 2020***). Epidemiological evidence also indicates that Δ9-THC exposure is associated with long-lasting adverse effects, with exposures in parents affecting the offspring (***Smith et al., 2020***; ***Szutorisz and Hurd, 2018***). Despite this accumulating evidence,

the molecular impact and mechanisms of Δ9-THC exposure at the earliest stages of germ cells development remain to be determined.

Progression through states of pluripotency is controlled by metabolic reprogramming in the early mammalian embryo (*Verdikt and Allard, 2021*; *Zhang et al., 2018*). Accordingly, cultured pluripotent stem cells (PSCs) exhibiting different developmental potentials are marked by specific metabolic signatures, similar to the ones displayed by their in vivo counterparts in the embryo. For instance, mouse embryonic stem cells (ESCs) are naïve PSCs that are functionally equivalent to the inner cell mass (ICM) of the E3.5 preimplantation mouse blastocyst (*Nichols and Smith, 2009*). The extended developmental potential of mouse ESCs is associated with their metabolic bivalence, as these cells rely on both glycolysis and oxidative phosphorylation for energy production. Differentiation of naïve ESCs into primed PSCs such as epiblast-like cells (EpiLCs) is accompanied by an important metabolic shift towards aerobic glycolysis, in link with a highly proliferative phenotype and a more restricted developmental potential (*Hayashi et al., 2017*; *Verdikt and Allard, 2021*; *Zhang et al., 2018*). Primordial germ cells (PGCs), the embryonic precursors of gametes in metazoans (*Hayashi et al., 2007*; *Kurimoto and Saitou, 2018*), are considered dormant totipotent cells because they possess the unique ability to reacquire totipotency upon fertilization (*Hayashi et al., 2017*). In the mouse embryo, the precursors to PGCs arise around embryonic day 6.25 (E6.25) from formative pluripotent cells in the epiblast (*Kinoshita et al., 2021*; *Kurimoto and Saitou, 2018*). Progressive increase in oxidative phosphorylation correlates with the specification and differentiation from epiblast PSCs towards PGCs, a process that can be replicated in vitro by inducing PGC-like cells (PGCLCs) from EpiLCs (*Hayashi et al., 2011*; *Yao et al., 2022*). In particular, the extensive metabolic, transcriptional, and epigenetic reprogramming that PGCs undergo during their development has been proposed to be uniquely sensitive to environmental insults, with potential consequences in the offspring (*Verdikt et al., 2022*).

Here, we deployed this in vitro differentiation system to investigate the impact of Δ9-THC exposure on early developmental stages. We demonstrate that exposure of ESCs and EpiLCs to Δ9-THC durably alters their metabolome. We reveal that, in the absence of continuous exposure, metabolic memory of Δ9-THC is passed onto the PGCLCs stage leading to transcriptional defects in these cells. Together, our findings highlight the role of metabolic reprogramming as a mechanism for early developmental Δ9-THC exposure.

## Results

### Δ9-THC induces cellular proliferation of mouse embryonic stem cells but not of mouse epiblast-like cells

To model the impact of early Δ9-THC exposure on early embryonic events, we first tested three distinct developmental windows: (1) exposure of ESCs, (2) exposure of EpiLCs, and (3) combined ESCs +EpiLCs exposure (*Figure 1A*). Cells were either exposed to the vehicle (mock) or exposed to Δ9-THC in a wide dose range of 0.1 nM-100µM, corresponding to the reported physiologically-relevant concentrations of Δ9-THC in cannabis users (*Fuchs Weizman et al., 2021*; *Hunault et al., 2009*; *Pacifici et al., 2020*).

The viability of ESCs exposed to increasing concentrations of Δ9-THC for 48 hr was not significantly altered until the maximal dose of 100 µM, subsequently serving as a positive control, at which only 13.17% of cells remained alive (*Figure 1B*, p=$5.10^{-15}$, unpaired T-test). While no significant changes in viability were observed between 10 nM and 1 µM Δ9-THC, the number of viable ESCs significantly increased by 1.69, 1.52 and 1.28-fold, respectively, compared to the mock-treated condition (*Figure 1C*, p=0.002, p=0.01 and p=0.03 for 10 nM, 100 nM and 1 µM of Δ9-THC, unpaired T-test). Of note, the number of viable ESCs significantly increased for doses of Δ9-THC as low as 1 nM (*Figure 1—figure supplement 1 and*, 0.59-fold increase, p=0.0005, unpaired T-test), corresponding to doses found in users of 'light cannabis' products (*Pacifici et al., 2020*). To determine whether the increased number of viable cells recovered after Δ9-THC exposure was due to higher proliferation, we performed bromodeoxyuridine (BrdU) labeling experiments. Exposed cells were pulsed with BrdU for 30 min, and its incorporation in actively dividing cells was measured by flow cytometry. The percentage of BrdU-positive ESCs significantly increased between 10 nM and 10 µM of Δ9-THC compared to the mock-treated condition (*Figure 1D*, p=0.01, p=0.001, p=0.05 and p=0.01 for 10 nM, 100 nM,

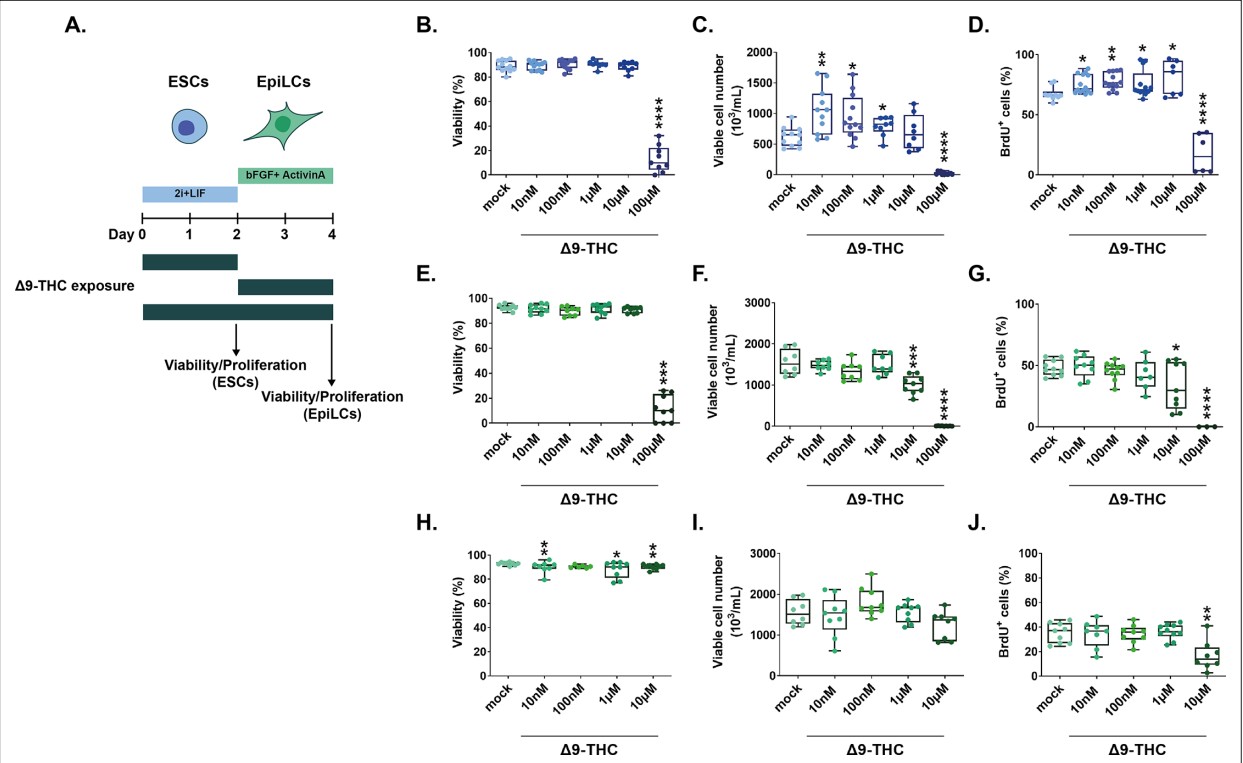

**Figure 1.** Δ9-THC exposure induces the proliferation of ESCs but not EpiLCs (**A**) Diagram illustrating Δ9-THC exposure scheme and experimental strategy. bFGF: basic fibroblast growth factor, ESCs, embryonic stem cells; EpiLCs, epiblast-like cells; LIF, leukemia inhibitory factor. (**B, E, H**) Whisker boxplot indicating the median cellular viability of stem cells exposed to the different Δ9-THC doses and associated errors. (**C, F, I**) Whisker boxplot indicating the median number of viable cells exposed to the different Δ9-THC doses indicated and associated errors. (**D, G, J**) Whisker boxplot indicating the median percentage of BrdU-stained cells exposed to the different Δ9-THC doses and associated errors. ESCs exposed cells are presented in (**B, C and D**). EpiLCs exposed cells deriving from unexposed ESCs are presented in (**E, F and G**). EpiLCs exposed cells deriving from exposed ESCs are presented in (**H, I and J**). At least three independent biological repeats with three technical replicates (N=3, n=3). Statistical significance: *(p<0.05), **(p<0.01), ***(p<0.001), ****(p<0.0001).

The online version of this article includes the following figure supplement(s) for figure 1:

**Figure supplement 1.** Δ9-THC induces ESCs proliferation for as low as 1 nM.

**Figure supplement 2.** Δ9-THC induces alteration in ESCs cell cycle.

**Figure supplement 3.** Δ9-THC exposure in male ESCs also provokes cell proliferation.

**Figure supplement 4.** hESCs cell number decreases upon Δ9-THC exposure.

1 μM and 10 μM of Δ9-THC, unpaired T-test). Co-staining with DAPI (4',6-diamidino-2-phenylindole) to analyze the cell cycle showed that, at 100 nM of Δ9-THC, a significantly higher proportion of cells were in the G2/M phase (*Figure 1—figure supplement 2 and*, 0.76-fold increase, p=0.02, unpaired T-test), consistent with cellular proliferation. Finally, whereas the present data was obtained using female ESCs, the proliferative effect of Δ9-THC exposure on ESCs was also observed in male ESCs (*Figure 1—figure supplement 3*), suggesting that proliferation is sex-independent.

Next, we derived EpiLCs from unexposed ESCs and performed the same dose-response experiments. Akin to ESCs, Δ9-THC exposure in EpiLCs did not significantly alter cellular viability until the dose of 100 μM (*Figure 1E*, 11.56% of viable cells, p=1.6$^{-13}$, unpaired T-test). However, contrary to ESCs, Δ9-THC exposure in EpiLCs did not significantly increase the number of viable cells nor the percentage of BrdU-positive cells (*Figure 1F* and *Figure 1G*). For the dose of 10 μM of Δ9-THC, viable EpiLCs numbers and BrdU-positive EpiLCs decreased compared to the mock-treated condition (*Figure 1F*, 1.53-fold decrease, p=0.0007, *Figure 1G*, 1.42-fold decrease, p=0.0024, unpaired T-test). When continuously exposing ESCs and EpiLCs, cell viability was more significantly and negatively impacted, except at the dose of 100 nM of Δ9-THC (*Figure 1H*). Deriving EpiLCs from exposed ESCs and exposing them to Δ9-THC for 48 hr did not significantly affect either their cell number nor their

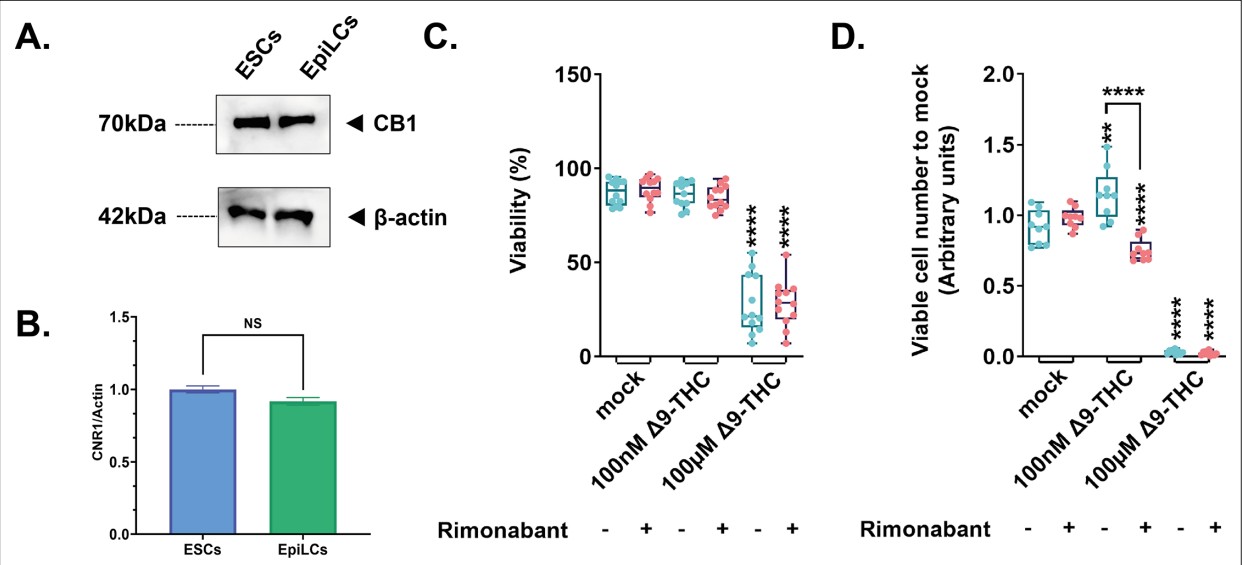

**Figure 2.** Implication of the CB1 receptor in the proliferative phenotype. (**A**) Western blot analysis of transmembrane protein extracts of ESCs or EpiLCs. Antibodies raised against CB1 or β-actin serving as a loading control were used for immunoblotting. (**B**) Quantification of the gel presented in (**A**) was done using Image Studio (version 5.2). (**C**) Whisker boxplot indicating the median cellular viability of stem cells exposed to the different Δ9-THC and rimonabant doses indicated and their associated errors. (**D**) The median numbers of viable cells exposed to the different Δ9-THC and rimonabant doses indicated were normalized to their own control (+/-rimonabant). Median and associated errors were plotted in whisker boxplots. At least three independent biological repeats with three technical replicates (N=3, n=3). Statistical significance: **(p<0.01), ****(p<0.0001).

The online version of this article includes the following source data for figure 2:

**Source data 1.** Western blot data for CB1 and β-actin Raw blots from *Figure 2A* are presented.

---

incorporation of BrdU (*Figure 1I* and *Figure 1J*), indicating that the increased proliferation observed at the ESCs stage is not carried through the naive-to-prime transition.

Finally, we further assessed the impact of Δ9-THC exposure in human pluripotent stem cells. Human embryonic stem cells (hESCs) differ from their murine counterparts, particularly because they resemble a primed pluripotent state, akin to mouse EpiLCs (*Weinberger et al., 2016*). Cell viability of hESCS continuously exposed to 100 nM of Δ9-THC was not significantly impacted (*Figure 1—figure supplement 4A*). However, hESCs number was significantly decreased upon Δ9-THC exposure (*Figure 1—figure supplements 4B and 1*.26-fold decrease, p=0.004, unpaired T-test), in a similar trend as observed with mouse EpiLCs (*Figure 1E* and *Figure 1I*).

Together, the systematic testing of different exposure schemes of Δ9-THC in different pluripotent stem cell populations (mouse and human ESCs and mouse EpiLCs) revealed that physiologically relevant doses of Δ9-THC (10 nM-1µM) specifically stimulate the proliferation of mouse ESCs, but not of human ESCs, nor of mouse EpiLCs, whether the latter are derived from exposed ESCs or not.

## Expression of the CB1 receptor does not explain differences in proliferative outcomes

We next sought to understand the source of variation in proliferative outcomes in response to Δ9-THC between naive mouse embryonic stem cells and primed pluripotent epiblast-like cells. Such differential effects have been previously reported, with Δ9-THC eliciting the proliferation of neural progenitors (*Galve-Roperh et al., 2013*) and of human breast carcinoma cell lines (*Takeda et al., 2008*) but suppressing the proliferation of activated CD4[+] T cells (*Yang et al., 2016*) and of non-small cell lung cancer cells (*Preet et al., 2008*). In these studies, the differential expression of cannabinoid receptors at the cell surface was proposed to primarily mediate the variation in cellular outcomes.

We therefore first tested whether expression levels of CB1 varied between ESCs and EpiLCs. Western-blot analysis of membrane proteins revealed however that CB1 was expressed at the same levels at the cell surface of both ESCs and EpiLCs (*Figure 2A and B*). We next determined whether the Δ9-THC-induced proliferative phenotype in ESCs was due to the engagement of the CB1 cannabinoid

receptor. To do so, ESCs were pretreated for 1 hr with 1 µM of SR141716 (also known as rimonabant, a specific CB1 blocker [*Rinaldi-Carmona et al., 1994*]) then exposed to 100 nM or 100 µM of Δ9-THC for 48 hr. Rimonabant pre-treatment did not significantly alter the viability of ESCs compared to conditions exposed to Δ9-THC only (*Figure 2C*) but abolished Δ9-THC-induced ESCs increased cell number at 100 nM Δ9-THC (*Figure 2D*, 1.53-fold decrease, p=1.66$^{-05}$, when comparing 100 nM of Δ9-THC+/-1 µM of SR141716, unpaired T-test). Notably, SR141716 pre-treatment, while not altering cell viability, reduced cell number compared to control, suggesting a basal role for CB1 in promoting proliferation.

Thus, the expression of CB1 at the cell surface does not explain the differential impact of Δ9-THC on ESC and EpiLC proliferation even if CB1 engagement is a required event for this effect in ESCs.

## Δ9-THC exposure increases glycolysis in ESCs and EpiLCs

In the central nervous system, Δ9-THC is a known metabolic perturbator which increases bioenergetic metabolism (*Bartova and Birmingham, 1976*; *Bénard et al., 2012*). As mentioned above, the transition of naïve ESCs into the primed state of EpiLCs is accompanied by a switch to glycolysis for energy production (*Hayashi et al., 2017*; *Verdikt and Allard, 2021*). Thus, to capture the impact of Δ9-THC at every point of their transition between metabolic states, we used the continuous exposure scheme of ESCs and EpiLCs outlined in *Figure 1H–J*. Similar to our other exposure schemes, at lower Δ9-THC doses, the proliferation of ESCs was observed but not of EpiLCs. We performed these exposures in a wide Δ9-THC dose range (10 nM-10µM) followed by bioenergetics assessment (*Figure 3A*).

First, we assessed the global energy metabolism of exposed cells by measuring the nicotinamide adenine dinucleotide (phosphate) couple ratios (NAD(P)+/NAD(P)H) using the WST-1 assay. In ESCs, the ratio of NAD(P)+/NAD(P)H significantly increased 1.57, 1.54, 1.29, and 1.38 -fold, for 10 nM, 100 nM, 1 µM, and 10 µM of Δ9-THC, respectively, compared to the mock-treated condition (*Figure 3B*, p=2.87$^{-06}$, p=8.01$^{-05}$, p=0.03, and p=0.0003 for 10 nM, 100 nM, 1 µM and 10 µM of Δ9-THC, unpaired T-test). In contrast, no significant increase was observed in NAD(P)+/NAD(P)H ratios in exposed EpiLCs (*Figure 3B*). Consistent with the impact of continuous Δ9-THC exposure

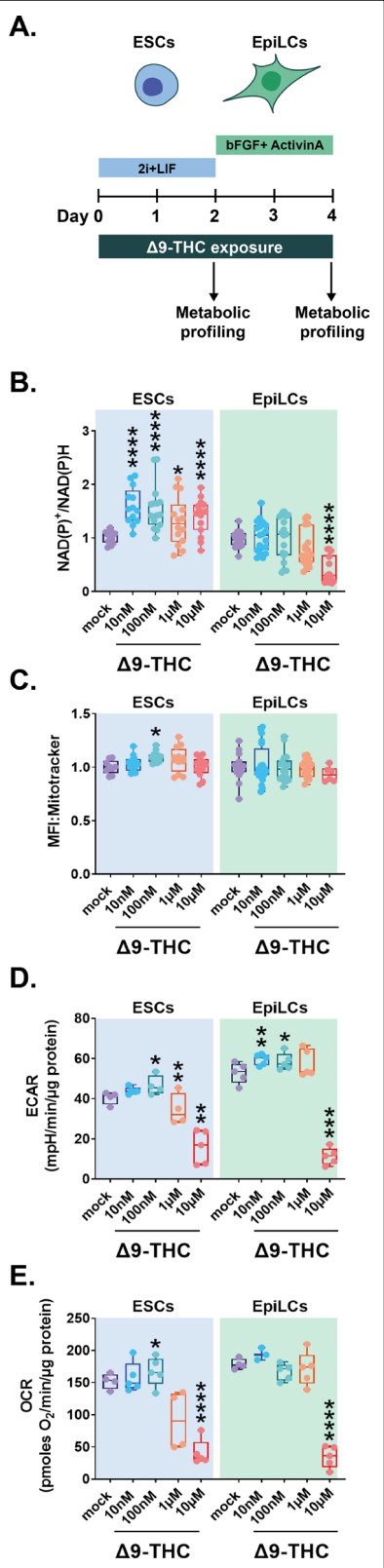

**Figure 3.** Δ9-THC exposure provokes an increase in glycolytic rates in ESCs and EpiLCs. (**A**) Diagram illustrating Δ9-THC exposure scheme and experimental strategy. (**B**) The NAD(P)+/NADPH ratio of stem

*Figure 3 continued on next page*

*Figure 3 continued*

cells exposed to the different Δ9-THC doses was normalized to the one measured in the mock-treated condition. Median and associated errors were plotted in whisker boxplots. (**C**) Mean fluorescence intensity (MFI) associated with the Mitotracker CMXRos stain was normalized to the one measured in the mock-treated condition. Median and associated errors were plotted in whisker boxplots. (**D**) Median and associated error of the maximal extracellular acidification rate (ECAR) measured in cells exposed to the different Δ9-THC doses and normalized to the protein content was plotted in whisker boxplots. (**E**) Median and associated error of the maximal oxygen consumption rate (OCR) measured in cells exposed to the different Δ9-THC doses and normalized to the protein content was plotted in whisker boxplots. For (**B and C**), 5 technical repeats of 3 biological repeats (n=15) were plotted. One representative experiment out of three independent experiments was used to plot results in (**D and E**). Statistical significance: *(p<0.05), **(p<0.01), ***(p<0.001), ****(p<0.0001).

The online version of this article includes the following figure supplement(s) for figure 3:

**Figure supplement 1.** hESCS metabolism is slightly but significantly impacted by Δ9-THC exposure.

**Figure supplement 2.** Extracellular acidification rates and oxygen consumption rates in ESCs and EpiLCs upon Δ9-THC exposure.

on EpiLCs viability (*Figure 1H*), the NAD(P)+/NAD(P)H ratios significantly decreased at 10 μM of Δ9-THC in EpiLCs (*Figure 3B*, 59% decrease for 10 μM of Δ9-THC compared to the mock-treated condition, p=6.55$^{-09}$, unpaired T-test). Of note, the NAD(P)+/NAD(P)H ratios were slightly but significantly decreased at 100 nM of Δ9-THC in hESCs (*Figure 3—figure supplement 1*, 0.09-fold decrease, p=0.04, unpaired T-test), in agreement with the deleterious effect of Δ9-THC on cell number (*Figure 1—figure supplement 4B*).

Because the elevated NAD(P)+/NAD(P)H levels in Δ9-THC-exposed mouse ESCs could indicate increased mitochondrial activity in the context of oxidative phosphorylation (*Locasale and Cantley, 2011*), we next studied changes in the mitochondrial membrane potential of exposed cells using the Mitotracker CMXRos fluorescent dye (*Pendergrass et al., 2004*). A significant increase in mean fluorescence intensity (MFI) associated with the mitochondrial stain was observed at 100 nM of Δ9-THC in ESCs (*Figure 3C*, p=0.02, unpaired T-test), indicating that, at this dose, the observed increase in NAD(P)+/NAD(P)H could be explained by higher mitochondrial membrane potential. By contrast, no change in EpiLCs' mitochondrial activity was detected (*Figure 3C*), consistent with these cells relying on glycolysis for energy production (*Hayashi et al., 2017*; *Verdikt and Allard, 2021*).

Changes in mitochondrial activity in ESCs upon Δ9-THC exposure, although significant, remained modest and are unlikely to be the sole contributor to the more significant increase in NAD(P)+/NAD(P)H upon exposure. Thus, we performed an in-depth analysis of the differential impact of Δ9-THC on ESCs and EpiLCs bioenergetics by measuring both glycolysis (extracellular acidification rate, ECAR) and mitochondrial respiration (oxygen consumption rate, OCR) using a Seahorse bioanalyzer. At 100 nM of Δ9-THC, the maximal glycolytic capacity of both ESCs and EpiLCs increased significantly (*Figure 3D*, 15% increase, p=0.03 and 22% increase, p=0.03 for ESCs and EpiLCs, respectively, compared to the mock-treated condition, unpaired T-test). In both cell types, a significant decrease in glycolytic capacity was observed at 10 μM of Δ9-THC (*Figure 3D*, 39.8% reduction, p=0.0006 and 44.8% reduction, p=0.0001, for ESCs and EpiLCs, respectively, compared to the mock-treated condition, unpaired T-test). Of note, the maximal glycolytic capacity of EpiLCs in the untreated condition was higher than the one of ESCs, in agreement with their metabolic shift towards aerobic glycolysis (*Figure 3D*, 7.88% higher ECAR rate in mock-treated EpiLCs compared to mock-treated ESCs, p=0.03, unpaired T-test). As a consequence, Δ9-THC exposure significantly impacted more glycolysis in EpiLCs than ESCs, both in basal capacity and upon mitochondrial inhibition by oligomycin (*Figure 3—figure supplement 2A and B*). In addition, at 100 nM of Δ9-THC, the maximal respiratory capacity of ESCs was significantly increased compared to the mock-treated condition (*Figure 3E*, 21.8% increase, p=0.03, unpaired T-test). This increase was observed only for the maximal respiratory capacity of ESCs, but not for basal respiration, nor for ATP-linked respiration (*Figure 3—figure supplement 2C*), suggesting that Δ9-THC impact on mitochondrial respiration does not support increased energetic production. In agreement with EpiLCs' metabolic shift towards a glycolytic phenotype, increasing doses of Δ9-THC did not alter their maximal respiratory capacity (*Figure 3E*), nor their global oxygen consumption rate (*Figure 3—figure supplement 2D*). In both cell types, a significant decrease in oxygen consumption rate was observed at 10 μM of Δ9-THC (*Figure 3E* and *Figure 3—figure supplement 2C* and *Figure 1D*).

Together, our analysis of cellular bioenergetics following Δ9-THC exposure showed an increased glycolytic rate in ESCs that was also observed in EpiLCs. However, the increased oxygen consumption and the associated increase in mitochondrial activity were observed only in ESCs following exposure to 100 nM of Δ9-THC, likely for the oxidization of the accumulating pyruvate generated from glycolysis.

## Δ9-THC-induced increase in glycolysis supports anabolism and ESC proliferation

Because our data indicated that the impact of Δ9-THC exposure on stem cells' bioenergetics did not result in greater ATP production, we next sought to characterize the global metabolic impact of Δ9-THC in these cells. ESCs and EpiLCs were continuously exposed to 100 nM Δ9-THC and intracellular metabolites were detected and quantified by mass spectrometry (*Figure 4A–E*). To explore the metabolic signatures in the different samples, we performed a global principal component analysis (PCA) (*Figure 4B*). All samples clustered in well-defined groups of replicates, both by cell type on the first principal component (accounting for 65.81% of the variation) and by Δ9-THC exposure on the second principal component (accounting for 20.83% of the variation). Of the 126 metabolites detected in ESCs, 39 were significantly upregulated (*Figure 4C* and *Figure 4—figure supplement 1A*) and only two metabolites – NAPDH and Adenine – were significantly downregulated. Of the 138 metabolites detected in EpiLCs, 95 were significantly upregulated (*Figure 4C* and *Figure 4—figure supplement 1B*) and only one metabolite – NAPDH – was significantly downregulated. In agreement with the PCA, the overlap of over-expressed metabolites in response to Δ9-THC exposure was important between the two stem cell populations (*Figure 4C*, accounting for 79.49% and 32.63% of all upregulated metabolites in ESCs and EpiLCs, respectively). The functional interpretation of the significantly upregulated metabolites confirmed the Δ9-THC-associated increase in energy metabolism in the two stem cell populations. Indeed, amongst the 25 metabolic pathways upregulated, pyruvate metabolism and glycolysis were detected in both ESCs and EpiLCs (*Figure 4D* and *Figure 4E*, respectively). Increased mitochondrial respiration was also seen in ESCs with the enrichment of (ubi)quinone metabolism, indicating an increased synthesis of ubiquinone that serves as an electron carrier in oxidative phosphorylation. Of note, metabolite measurements showed that the ratio of glutathione in its reduced to oxidated form (GSH/GSSG) was unchanged in both stem cell types in response to Δ9-THC (*Figure 4—figure supplement 1C*), suggesting that the increased mitochondrial respiration does not cause an overt elevation of oxidative stress. Importantly, and in agreement with the PCA, in both ESCs and EpiLCs, Δ9-THC exposure elicited an increase in metabolic pathways that feed anabolic reactions, in particular contributing to the synthesis of amino acids (tyrosine, tryptophan, arginine, alanine, valine, (iso)leucine, etc.), nucleotides ('Pyrimidine metabolism', 'Purine metabolism'), NAD(P)+ ('Nicotinate and nicotinamide metabolism') and fatty acids ('Butanoate metabolism'; *Figure 4D* and *Figure 4E*).

Extensive metabolic profiling of ESCs and EpiLCs upon Δ9-THC exposure thus indicated that the increased glycolytic rates in both stem cell populations, rather than provoking an increased production of energy under the form of ATP, participated in increased anabolism. Such increased anabolism could explain the proliferation observed in ESCs upon Δ9-THC exposure. To test this hypothesis, we exposed ESCs to 100 nM of Δ9-THC for 48 hr as above but 24 hr before the harvest, cells were exposed to 10 mM of 2-deoxyglucose (2-DG), an inhibitor of glycolysis (*Barban and Schulze, 1961*). Despite increasing the energy stress (*Figure 4—figure supplement 2*), inhibition of glycolysis by 2-DG did not significantly impact viability over this shorter time frame and at this concentration (*Figure 4F*). Importantly, glycolytic inhibition by 2-DG abrogated the Δ9-THC-induced increase in both cell number and NAD(P)+/NAD(P)H levels (*Figure 4G* and *Figure 4H*, 1.39-fold reduction and $p = 5.56^{-05}$ and 1.68-fold reduction and $p = 0.0064$, respectively, when comparing 100 nM of Δ9-THC+/- 10 mM 2-DG, unpaired T-test). Thus, exposure to Δ9-THC increases anabolism in both ESCs and EpiLCs, however, this increased anabolism only supports cellular proliferation in ESCs.

## Δ9-THC exposure is associated with the upregulation of genes involved in anabolic pathways in ESCs but not in EpiLCs

Our data shows that Δ9-THC exposure increases anabolic pathways in both ESCs and EpiLCs and that this causes the proliferation of ESCs but not of EpiLCs. We thus next examined whether this differential impact of Δ9-THC on ESCs and EpiLCs was mirrored by a change in these cells' transcriptomes.

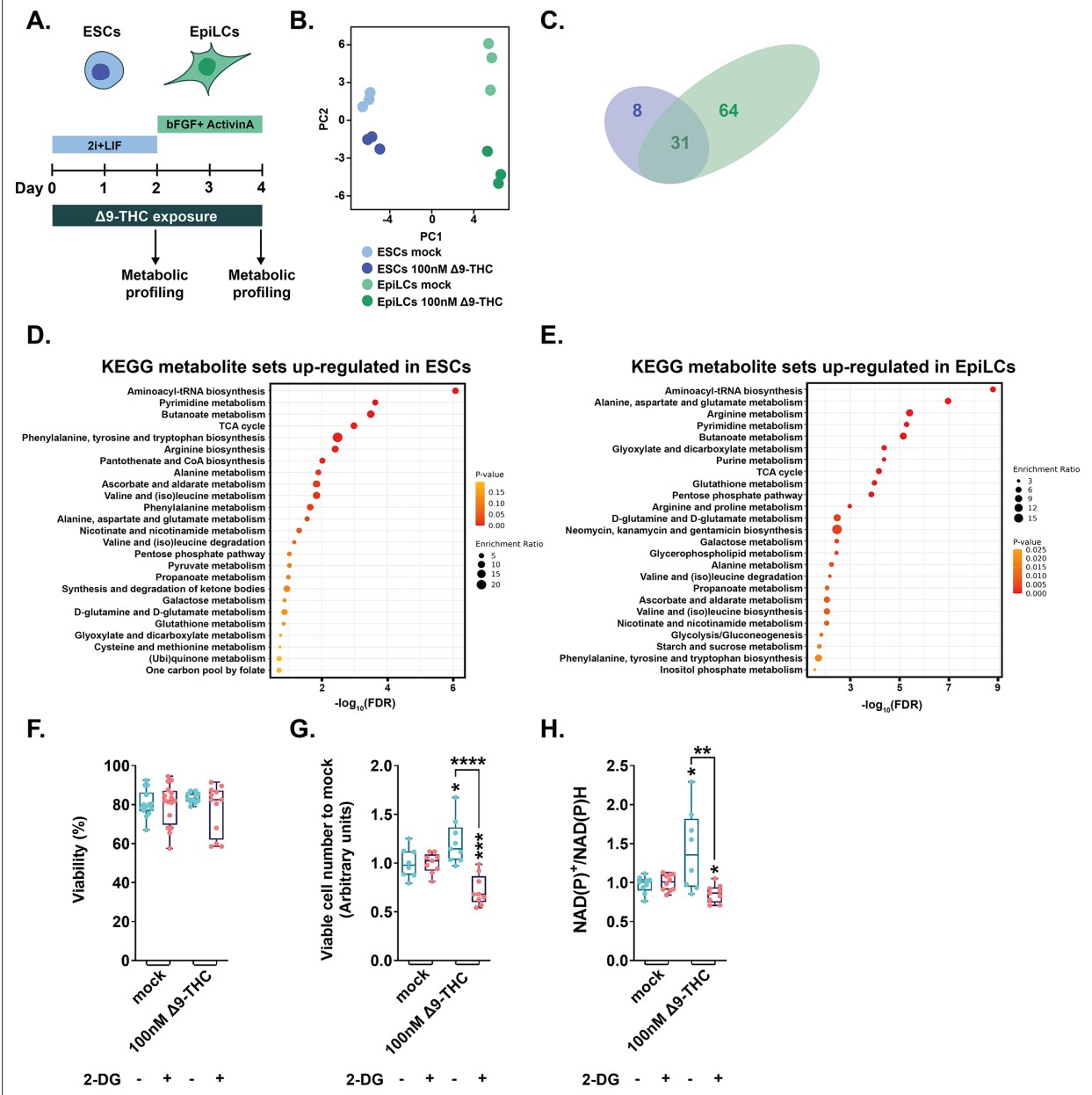

**Figure 4.** Δ9-THC-induced glycolysis sustain anabolism and ESCs proliferation (**A**) Diagram illustrating Δ9-THC exposure scheme and experimental strategy. (**B**) PCA of the metabolomics profiling of either ESCs or EpiLCs mock-exposed or exposed to 100 nM Δ9-THC. (**C**) Venn diagram showing the overlap in upregulated metabolites following Δ9-THC exposure in ESCs and EpiLCs. (**D and E**) KEGG metabolite sets enrichment analysis for upregulated metabolites in ESCs and EpiLCs, respectively, performed by MetaboAnalyst (*Pang et al., 2022*). *KEGG*, Kyoto Encyclopedia of Genes and Genomes. (**F**) Whisker boxplot indicating the median cellular viability of stem cells exposed to 100 nM of Δ9-THC and 10 mM of 2-DG, as indicated, and their associated errors. (**G**) The median numbers of viable cells exposed to 100 nM of Δ9-THC and 10 mM of 2-DG, as indicated, were normalized to their own control (+/-2 DG). Median and associated errors were plotted in whisker boxplots. (**H**) The NAD(P)+/NADPH ratio of stem cells exposed to 100 nM of Δ9-THC and 10 mM of 2-DG, as indicated, was normalized to the one measured in the mock-treated condition (+/-2 DG). Median and associated errors were plotted in whisker boxplots. At least three independent biological repeats with three technical replicates (N=3, n=3). Statistical significance: *(p<0.05), **(p<0.01), ***(p<0.001), ****(p<0.0001).

The online version of this article includes the following figure supplement(s) for figure 4:

**Figure supplement 1.** Metabolite profiling in ESCs and EpiLCs upon Δ9-THC exposure.

**Figure supplement 2.** Extracellular acidification rates and oxygen consumption rates in ESCs upon Δ9-THC and 2-DG exposure.

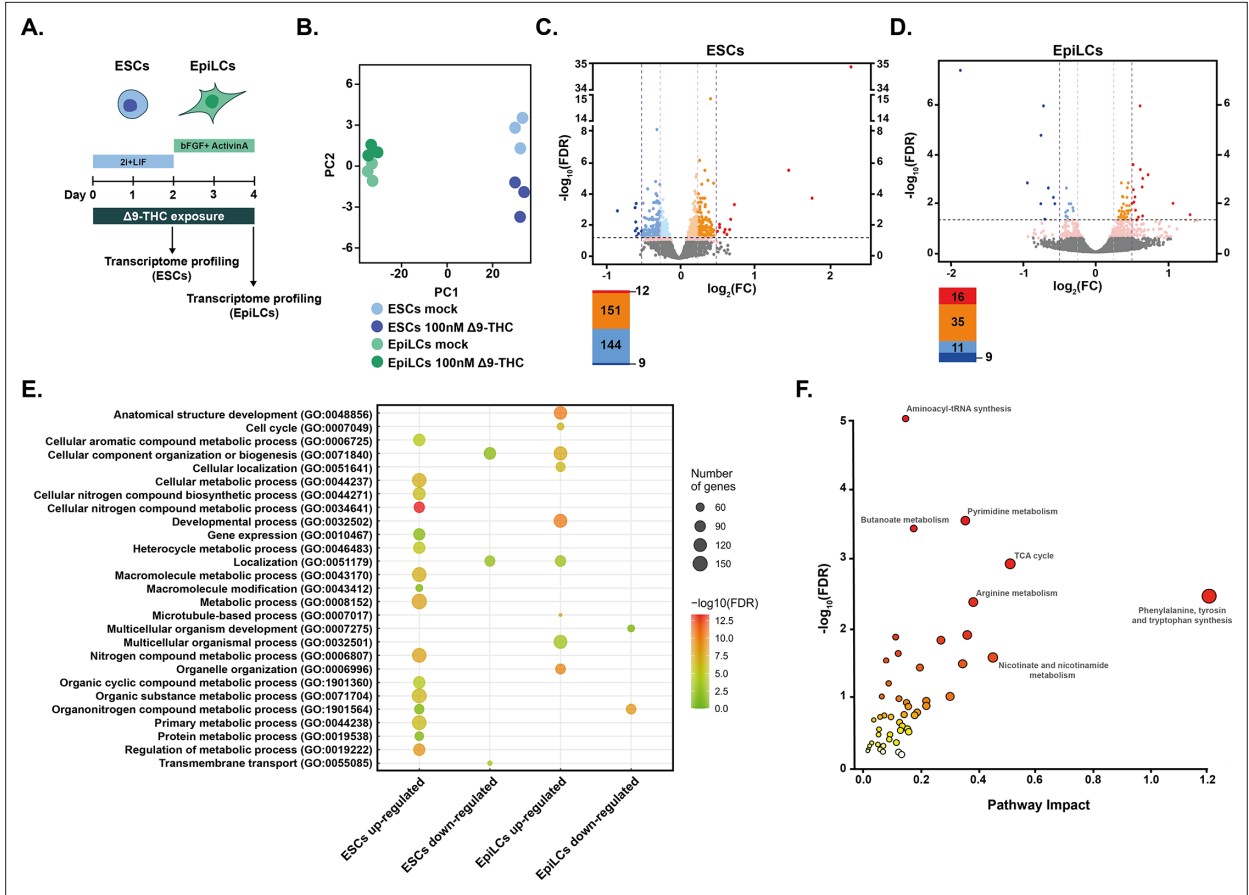

**Figure 5.** Metabolic changes following Δ9-THC exposure in ESCs are transcriptionally encoded. (**A**) Diagram illustrating Δ9-THC exposure scheme and experimental strategy. (**B**) PCA of the transcriptomics profiling of either ESCs or EpiLCs mock-exposed or exposed to 100 nM Δ9-THC. (**C and D**) Volcano plot in ESCs and EpiLCs, respectively, showing significance [expressed in $\log_{10}$(adjusted p-value or false-discovery rate, FDR)] versus fold-change [expressed in log2(fold-change, FC)]. Thresholds for significance (adjusted p-value ≤0.05) and gene expression fold-change [|(log2(FC)|>0.25 or |log2(FC)|>0.5] are shown as dashed lines. Color code is as follows: log2(FC) >0.5 in red, log2(FC) >0.25 in orange, log2(FC) >0 in light orange, log2(FC) <0 in light blue, log2(FC) >−0.25 in blue, log2(FC) >0.5 in dark blue and p-value <0.01 in pink. (**E**) Gene ontology (GO) terms associated with up- and downregulated DEGs [|(log2(FC)|>0.25 and p<0.01)] in ESCs and EpiLCs as determined by g:Profiler (***Raudvere et al., 2019***). (**F**) Joint pathway analysis performed by the multi-omics integration tool of MetaboAnalyst (***Pang et al., 2022***). The p-values were weighted based on the proportions of genes and metabolites at the individual pathway level.

The online version of this article includes the following figure supplement(s) for figure 5:

**Figure supplement 1.** Δ9-THC exposure does not alter markers of pluripotency.

**Figure supplement 2.** Δ9-THC exposure alters the expression of some epigenetic modifiers.

To this aim, we performed RNA-sequencing (RNA-seq) on ESCs and EpiLCs continuously exposed to 100 nM Δ9-THC or to the vehicle control (***Figure 5A***).

Unsupervised exploration of the global transcriptome by PCA revealed that the vast majority of data variation could be attributed to the cell type (PC1, accounting for 98% of the variation) rather than to Δ9-THC exposure (PC2, accounting for 1% of the variation, ***Figure 5B***). In addition, mining for the expression of pluripotency markers after Δ9-THC exposure suggested that Δ9-THC does not appear to influence the differentiation dynamics from ESCs to EpiLCs (***Figure 5—figure supplement 1***). We identified a low number of differentially expressed genes (DEGs) in both ESCs and EpiLCs (***Figure 5C*** and ***Figure 5D***, respectively), indicating that Δ9-THC exposure only moderately impacts ESC and EpiLC transcriptomes. In ESCs, only 12 genes were significantly upregulated with a log2(fold-change) >0.5 and only 9 were significantly downregulated at the same threshold (***Figure 5C***, significance corresponds to adjusted *P*-value ≤0.05). More genes were differentially expressed when looking at lower fold-changes (|log2(fold-change)|>0.25, ***Figure 5C***), confirming that the magnitude of transcriptional effects due to Δ9-THC exposure is moderate. This low transcriptional impact following

Δ9-THC exposure was also observed in EpiLCs (*Figure 5D*). Nevertheless, gene ontologies (GO) associated with Δ9-THC-induced DEGs revealed the biological significance of these low transcriptional changes (*Figure 5E*). In particular, GO terms associated with metabolic pathways involved in anabolism were significantly over-represented for upregulated genes in ESCs following Δ9-THC exposure (*Figure 5E*), such as: 'Cellular aromatic compound metabolic process', 'Cellular nitrogen compound biosynthetic process', 'Organonitrogen compound metabolic process'. This suggests that the glycolytic rewiring elicited by Δ9-THC exposure in ESCs has some transcriptional support. Indeed, when performing joint pathway integration between our transcriptomics data and our targeted metabolomics (*Pang et al., 2022*), we observed that Δ9-THC-induced perturbed genes and metabolites were associated with the observed anabolic effects (*Figure 5F*). In contrast, GO terms associated with metabolism were not found within the upregulated DEGs in EpiLCs. However, several GO terms relating to alterations in cellular components were enriched in EpiLCs (*Figure 5E*), such as: 'Organelle organization', 'Cellular component organization or biogenesis', 'Microtubule-based process'. This indicates that Δ9-THC exposure significantly upregulated genes in EpiLCs that impact organelles structure, integrity and position, in agreement with several reports in the literature (*Lojpur et al., 2019*; *Miller et al., 2019*).

Multiple reports indicate that Δ9-THC exposure alters the epigenome of sperm and brain tissue, both in terms of DNA methylation level and histone post-translational modifications (*Murphy et al., 2018*; *Prini et al., 2018*; *Schrott and Murphy, 2020*; *Watson et al., 2015*). Despite the low transcriptional impact of Δ9-THC exposure in ESCs and EpiLCs, we evaluated the expression of more than 100 genes encoding epigenetic modifiers in our RNA-seq datasets (*Supplementary file 1*). Data shows that the expression levels of multiple epigenetic modifiers were significantly altered, albeit at a low level, in either ESCs or EpiLCs upon Δ9-THC exposure (|log2(fold-change)|>0.25, *Figure 5—figure supplement 2*). In particular, the expression levels of the DNA dioxygenase *Tet2* were significantly increased in EpiLCs following Δ9-THC exposure, which would contribute to changes in DNA methylation dynamics and differentiation potential (*Sohni et al., 2015*). The expression of the two histone deacetylases *Hdac5* and *Hdac11* were significantly decreased in ESCs following Δ9-THC exposure, while the expression of the histone phosphorylase *Rps6ka5* increased under the same conditions. In EpiLCs, the expression of the histone methyltransferase *Kmt2c* and of the regulators of histone ubiquitination *Dzip3* and *Mysm1* were all significantly increased upon Δ9-THC exposure. Collectively, RNA-seq data suggests the existence of an epigenetic remodelling following Δ9-THC exposure, although further analysis of the respective epigenetic marks associated with these modifiers is needed.

Together, our analysis of ESCs and EpiLCs transcriptomes reveals a difference in the response of these stem cell populations to Δ9-THC exposure: the transcriptional alterations observed in ESCs supported their increased anabolism and proliferation, whereas changes in EpiLCs gene expression did not correlate with their metabolic changes.

## Proliferation of primordial germ cell-like cells stemming from prior Δ9-THC exposure

PGCs display a distinct transcriptomic and metabolic profile compared to their cellular precursors that are recapitulated in vitro during the differentiation of ESCs into EpiLCs and then of EpiLCs into PGCLCs. Thus, we asked whether the metabolic alterations observed in ESCs and EpiLCs could lead to an altered differentiation program in PGCLCs. To this aim, we continuously exposed ESCs and EpiLCs to a Δ9-THC dose range of 1 nM-1µM (or mock control), before changing to a Δ9-THC-free media and inducing PGCLCs differentiation (*Figure 6A*). In particular, we took advantage of ESCs that harbor two fluorescent reporters for germline markers, Blimp1:mVenus and Stella:CFP (*Ohinata et al., 2008*). Thus, the induction efficiency of PGCLCs within 5 days embryoid bodies can be detected by monitoring the fluorescence associated with each cell in flow cytometry, allowing for the determination of a double-negative population (DN), a single-positive population (SP) wherein Blimp1:mVenus is expressed and a double-positive population (DP) expressing both Blimp1:mVenus and Stella:CFP, which represents the true specified PGCLC population (*Figure 6—figure supplement 1*).

We first measured the impact of ESCs +EpiLCs Δ9-THC exposure on PGCLC induction efficiency. Flow analyses revealed a dose-dependent increase in the induction efficiency of SP and DP cell populations (*Figure 6B*). Specifically, at 100 nM Δ9-THC, a significant decrease in DN was observed, with a corresponding significant increase of 1.14-fold in SP and of 1.64-fold in DP cells (*Figure 6C*, p=0.0002,

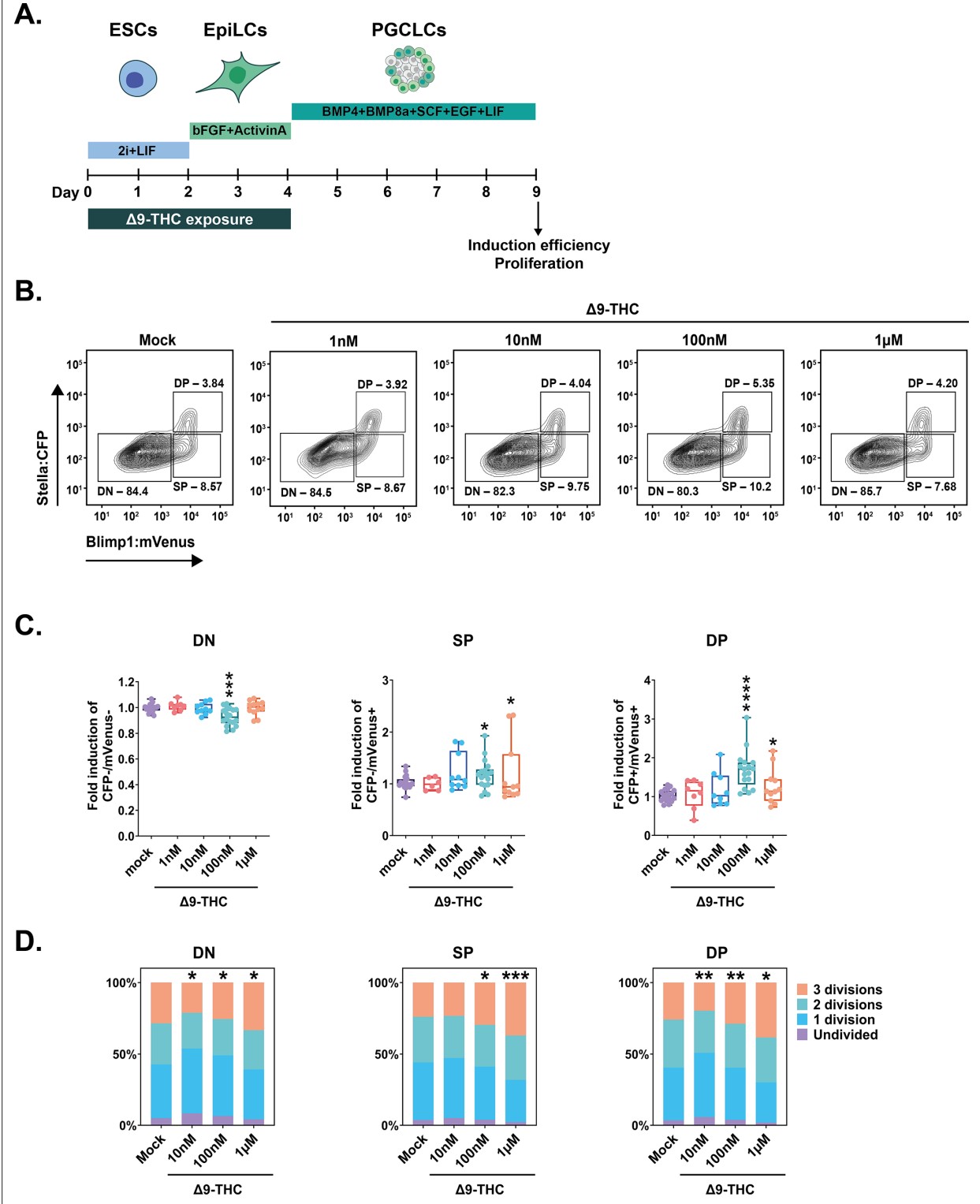

**Figure 6.** PGCLCs deriving from ESCs and EpiLCs exposed to 100 nM of Δ9-THC proliferate. (**A**) Diagram illustrating Δ9-THC exposure scheme and experimental strategy. (**B**) Representative flow contour plots showing distribution of live-gated events, gating strategy for Stella:CFP versus Blimp1:mVenus and percentages of cells in each subpopulations for ESCs and EpiLCs exposed to the different doses of Δ9-THC indicated. DN: double negative, SP: single positive, DP: double positive subpopulations. (**C**) The percentage of events in the gates associated to each subpopulation was normalized to the one measured in the mock-treated condition. Median and associated errors were plotted in whisker boxplots independently for each subpopulation. (**D**) Histograms showing CellTrace Yellow staining profile of cells arising from ESCs and EpiLCs exposed to the different doses of Δ9-THC

*Figure 6 continued on next page*

*Figure 6 continued*

indicated. The Y-axis represents the average percentage of cells in each category of subpopulations undividing (purple), undergoing 1 division (blue), 2 divisions (green) or 3 divisions (orange). One representative experiment out of three is represented. Statistical significance: *(p<0.05), **(p<0.01), ***(p<0.001), ****(p<0.0001).

The online version of this article includes the following figure supplement(s) for figure 6:

**Figure supplement 1.** PGCLCs gating and sorting strategy.

**Figure supplement 2.** Male PGCLCs deriving from ESCs and EpiLCs exposed to 100 nM of Δ9-THC proliferate.

**Figure supplement 3.** No residual Δ9-THC is detected in day 5 embryoid bodies.

p=0.05, and p=1.55⁻⁰⁶ for 100 nM of Δ9-THC in DN, SP, and DP populations respectively compared to the mock-treated condition, unpaired T-test). The same pattern was observed when male ESCs and EpiLCs exposed to 100 nM of Δ9-THC were differentiated in PGCLCs (*Figure 6—figure supplement 2*).

To determine if the increased proportion of PGCLCs generated from exposed precursors was due to higher proliferative kinetics, we performed a proliferation tracing assay (*Tempany et al., 2018*). The tracing dye was added to the cells on the day of aggregate formation, and fluorescence attenuation due to cell division was measured in each subpopulation on day 5. At 100 nM Δ9-THC, a smaller proportion of DN cells underwent two or three mitotic divisions compared to the control (*Figure 6D*, 1.14-fold fewer cells and 1.12-fold fewer cells, p=0.05 and p=0.04 for 2 divisions and 3 divisions, respectively, unpaired T-test). In parallel, for the same dose, a significantly higher proportion of SP and DP cells underwent three mitotic divisions compared to the control (*Figure 6D*, 1.24-fold and 1.11-fold, p=0.03 and p=0.0035, for 3 divisions in SP and DP cells, compared to the control, unpaired T-test). These results, therefore, indicate that the higher number of PGCLCs observed upon Δ9-THC exposure originates from their increased proliferation during their specification and differentiation.

Finally, we sought to determine if the increased PGCLC proliferation was not due to residual intracellular Δ9-THC persisting from EpiLCs over the span of PGCLCs differentiation. To do so, intracellular levels of Δ9-THC were quantified by mass spectrometry in the cells on the day of aggregate formation and in day 5 embryoid bodies (referred as to 'EpiLCs' and 'PGCLCs', respectively, in *Figure 6—figure supplement 3*). Data shows that no Δ9-THC could be detected in day 5 embryoid bodies, indicating that Δ9-THC does not persist to levels higher than the limit of detection of mass spectrometry (1 ng/mL). This suggests that the proliferative effects are not due to residual Δ9-THC persisting in the cells during differentiation towards PGCLCs. Thus, Δ9-THC causes an alteration of the developmental kinetics that PGCLCs normally undergo, even in the absence of direct continuous exposure.

## PGCLCs derived from Δ9-THC-exposed cells present an altered metabolism and transcriptome

Since exposure to Δ9-THC prior to their specification increased the number of PGCLCs and ESCs and PGCLCs share similar metabolic programs (*Hayashi et al., 2017*; *Verdikt and Allard, 2021*), we next sought to characterize their associated metabolic and transcriptional changes. We, therefore, assessed the impact of exposure of ESCs +EpiLCs to 100 nM Δ9-THC on PGCLCs metabolism (*Figure 7A*).

First, NAD(P)+/NAD(P)H assessment revealed a modest but significant 1.17-fold increase in NAD(P)+/NAD(P)H ratio in whole day 5 embryoid bodies deriving from exposed ESCs +EpiLCs compared to those deriving from mock-treated cells (*Figure 7B*, P=0.01, unpaired T-test). To garner cell type-specific information on whether these metabolic changes were related to mitochondrial activity and the differentiation of PGCLCs, we assessed the mitochondrial membrane potential of each subpopulation in day 5 embryoid bodies. Embryoid bodies were incubated with Mitotracker CMXRos (*Pendergrass et al., 2004*), and dissociated and analyzed by flow cytometry. The MFI associated with the mitochondrial stain was then measured in each subpopulation (*Figure 7C*). A significant increase in MFI was observed in DN, SP and DP populations deriving from exposed ESCs +EpiLCs compared to those deriving from mock controls (*Figure 7C*, 1.17, 1.16, 1.23 -fold, p=0.006, p=0.05 and p=0.01 for DN, SP, and DP, respectively, unpaired T-test). These results indicate that the metabolic changes induced by Δ9-THC prior to PGCLCs induction and differentiation are not reset during the profound reprogramming that PGCLCs undergo.

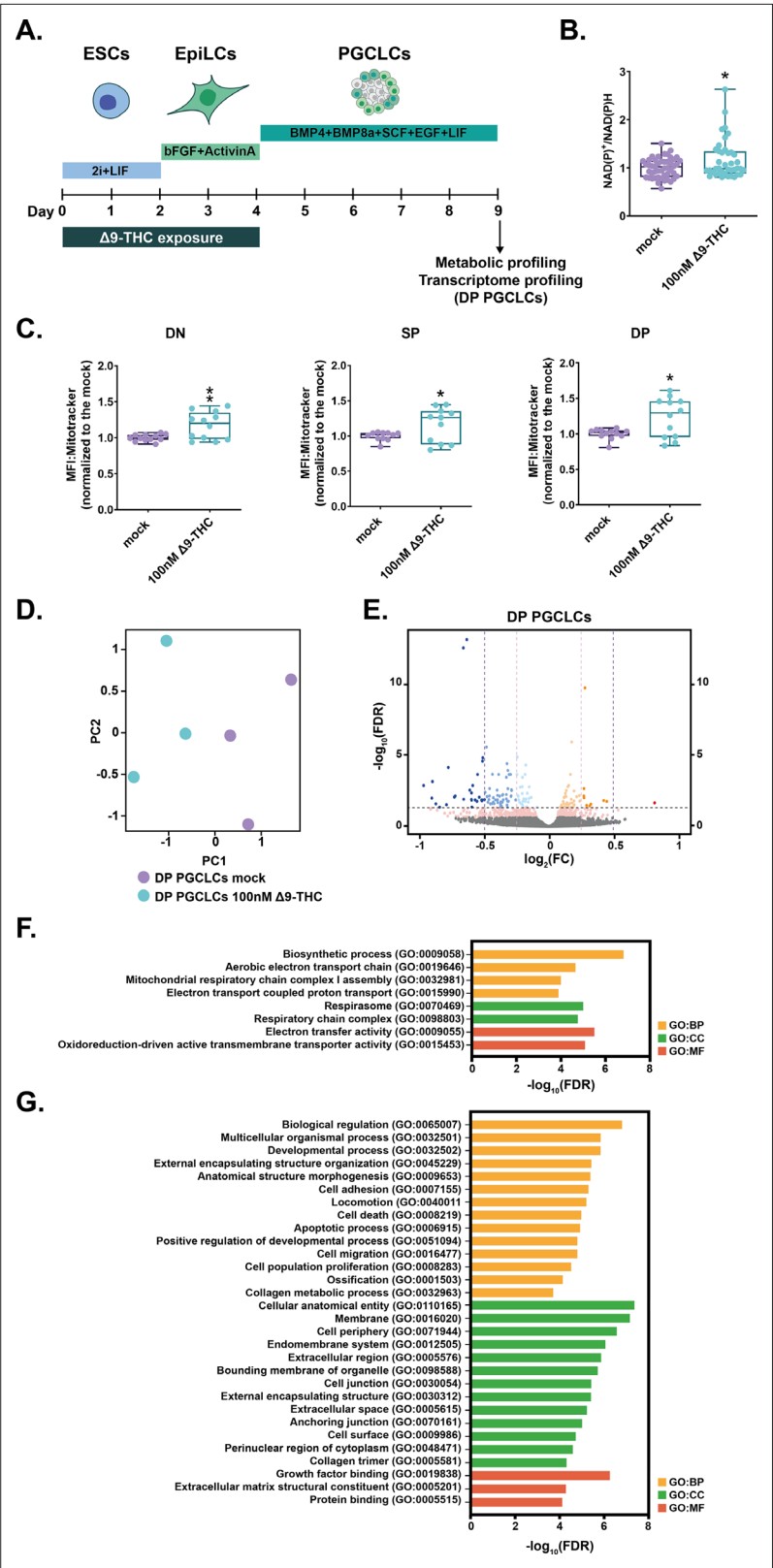

**Figure 7.** Δ9-THC exposure prior to specification increases mitochondrial respiration in PGCLCs. (**A**) Diagram illustrating Δ9-THC exposure scheme and experimental strategy. (**B**) The NAD(P)+/NADPH ratio of embryoid bodies arising from ESCs and EpiLCs exposed to 100 nM of Δ9-THC was normalized to the one measured in the mock-treated condition. Median and associated errors were plotted in whisker boxplot. (**C**) Mean fluorescence

*Figure 7 continued on next page*

*Figure 7 continued*

intensity (MFI) associated with the Mitotracker CMXRos stain in each subpopulation was normalized to the one measured in the mock-treated condition. Median and associated errors were plotted in whisker boxplots. (**D**) PCA of the transcriptomics profiling of DP PGCLCs deriving from ESCs and EpiLCs either mock-exposed or exposed to 100 nM Δ9-THC. (**E**) Volcano plot in DP PGCLCs showing significance [expressed in $\log_{10}$(adjusted p-value or false-discovery rate, FDR)] versus fold-change [expressed in log2(fold-change, FC)]. Thresholds for significance and different enrichment ratios [|(log2(FC)|>0.25 or |log2(FC)|>0.5)] are shown as dashed lines. Color code is as follows: log2(FC) >0.5 in red, log2(FC) >0.25 in orange, log2(FC) >0 in light orange, log2(FC) <0 in light blue, log2(FC) >−0.25 in blue, log2(FC) >0.5 in dark blue and p-value <0.01 in pink. (**F and G**) Gene ontology (GO) terms associated with up- and downregulated DEGs [|(log2(FC)|>0.25 and p<0.01)], respectively, as determined by g:Profiler (*Raudvere et al., 2019*). Statistical significance: *(p<0.05), **(p<0.01).

Because our results indicated a sustained impact of Δ9-THC beyond the period of direct exposure, we further examined PGCLCs by performing a transcriptomic analysis. In particular, day 5 embryoid bodies deriving from ESCs +EpiLCs, either exposed to 100 nM of Δ9-THC or mock-exposed, were sorted and the total RNA of DP subpopulations, representing true PGCLCs, was analyzed by RNA-seq (*Figure 7A*). Unsupervised analysis of the global transcriptome in DP PGCLCs by PCA delineated a transcriptional signature of prior Δ9-THC exposure (*Figure 7D*, PC1 accounting for 59% of the variance and PC2 accounting for 24% of variance). Volcano plot of DEGs between DP PGCLCs deriving from mock- or 100 nM Δ9-THC-exposed ESCs and EpiLCs revealed that most of the significant transcriptional change was towards downregulation rather than upregulation (*Figure 7E*, 11 genes were significantly upregulated whereas 97 were significantly downregulated, |log2(fold-change)|>0.25 and adjusted p-value ≤0.05). Despite the low number of upregulated DEGs, the functional annotation of their associated GO terms showed that all terms enriched corresponded to metabolic processes involved in oxidative phosphorylation (*Figure 7F*, 'Aerobic electron transport chain', 'Mitochondrial respiratory chain complex I assembly', 'Electron transport couple proton transport'). Thus, our data indicate that the metabolic changes induced by exposure to Δ9-THC prior to PGCLCs specification are retained through transcriptional reprogramming. Importantly, while our results show that pre-specification Δ9-THC exposure increases PGCLCs number and mitochondrial activity, the functional annotation of GO terms associated with downregulated DEGs suggests degradation of PGCLCs quality. Indeed, and reminiscent of GO terms observed in EpiLCs, several GO terms relating to alterations in structural cellular components ('Anatomical structure morphogenesis', 'Cellular anatomical entity'), and in particular the interface with the extracellular environment ('External encapsulating structure organization', 'Membrane', 'Cell periphery', 'Extracellular region', 'Extracellular space', 'Extracellular matrix structural constituent') were enriched (*Figure 7G*). Furthermore, GO terms associated with cell adhesion and junction ('Cell adhesion', 'Cell migration', 'Collagen metabolic process', 'Cell junction', 'Anchoring junction', 'Collagen trimer') were also enriched in downregulated genes.

Together, our data show that Δ9-THC exposure in ESCs and EpiLCs durably alters their metabolome and that these changes are carried through PGCLCs specification and differentiation, leading to an alteration of PGCLCs transcriptional program (*Figure 8*).

## Discussion

With greater social acceptance and legalization, cannabis use has increased worldwide (*Mennis et al., 2023*; *U.N Office on Drugs and Crime, 2022*). Yet, the impact of such heightened use on reproductive functions, and in particular, on the earliest developmental stages is not well understood. Cannabis use directly alters adult male fertility and causes abnormal embryo implantation (*Lo et al., 2022*). Using a well-characterized in vitro model of early embryonic differentiation events culminating in the differentiation of PGCLCs, our study is the first to shed light on the impact of Δ9-THC at these stages which unfold during the first trimester in humans (*Chabarria et al., 2016*; *Volkow et al., 2019*; *Young-Wolff et al., 2018*).

Our data revealed the differential effects of Δ9-THC on naïve and primed pluripotent stem cells, respectively represented by mouse ESCs and EpiLCs. In particular, exposure to Δ9-THC increased ESC proliferation which was in a similar range to what has been previously reported for human breast carcinoma cell lines (about 30–50% between 10 nM and 1 μM of Δ9-THC) (*Takeda et al., 2008*). Differential expression and use of cannabinoid receptors on the surface of exposed cells have been shown

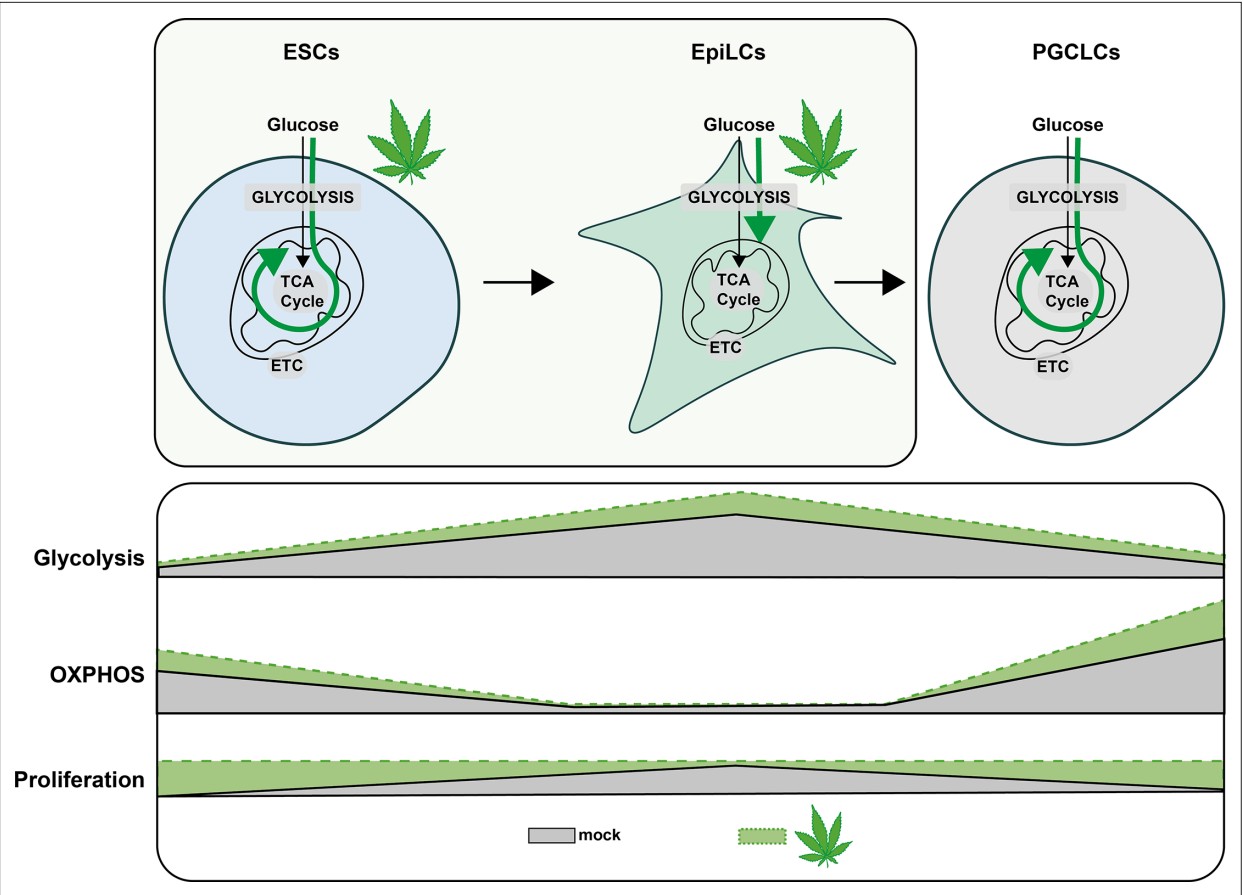

**Figure 8.** Metabolic impact of Δ9-THC exposure in pluripotent stem cells and primordial germ cells-like cells. Diagram illustrating the impact of Δ9-THC exposure on stem cells metabolism.

to correlate with Δ9-THC proliferative phenotypes (*Galve-Roperh et al., 2013*; *Preet et al., 2008*; *Takeda et al., 2008*; *Yang et al., 2016*). However, our experiments demonstrated that despite being required for Δ9-THC-induced proliferation in ESCs, CB1 expression did not significantly differ at the surface of ESCs and EpiLCs.

Because Δ9-THC is a known perturbator of mitochondrial function as previously described in the central nervous system (*Bartova and Birmingham, 1976*; *Bénard et al., 2012*), we studied the metabolic impact of its exposure in ESCs and EpiLCs. Our data indicate that, at 100 nM, Δ9-THC exposure increased the glycolytic rate in both ESCs and EpiLCs. Bioenergetics analyses and metabolite measurements showed that this increased glucose metabolism did not support increased energy production in the mitochondria, but rather, that it led to the accumulation of metabolic intermediates used in anabolic reactions for the synthesis of amino acids, nucleotides, and lipids. Thus, the metabolic signatures associated with Δ9-THC exposure are reminiscent of those inherently occurring during naïve-to-prime transition, during which increased aerobic glycolytic rates feed anabolic reactions ultimately fueling proliferation (*Lunt and Vander Heiden, 2011*). We verified this model by testing the requirement of increased glycolysis to support proliferation and indeed observed that ESCs proliferation upon Δ9-THC exposure is abrogated in the presence of the glycolytic inhibitor 2-DG.

Transcriptomic analyses revealed that the metabolic reprogramming induced by Δ9-THC exposure in ESCs was transcriptionally encoded, with increased expression of genes involved in anabolic pathways. In contrast, functional annotations of DEGs in EpiLCs did not show such transcriptional control of increased anabolism. Comparing the outputs of the metabolomic and transcriptomic analyses (i.e. PCA plots and volcano plots), the impact of Δ9-THC at these early stages seems to be primarily metabolic, although the moderate effects on the transcriptome appear to support the metabolic outcome as revealed by our integrated analysis (*Figure 5F*). Together, we propose that Δ9-THC exposure elicits

a reprogramming of ESCs that (1) coaxes them to rely more on aerobic glycolysis, (2) drives anabolic pathways, and therefore (3) leads to their proliferation. In EpiLCs, the impact of Δ9-THC exposure is not sufficient to override the cellular and metabolic programs of these already highly proliferative cells that are fully reliant on aerobic glycolysis (*Figure 8*).

Finally, we assessed the impact of Δ9-THC exposure in ESCs and EpiLCs on the differentiation of PGCLCs. Our data indicate that at the physiologically relevant dose of 100 nM of Δ9-THC, a significant increase in PGCLCs was observed. In particular, during PGCLCs differentiation, metabolic reprogramming and increased oxidative phosphorylation play a critical role in the reacquisition of an extended developmental potential (*Hayashi et al., 2017*; *Verdikt and Allard, 2021*). Thus, we investigated whether the metabolic alterations observed in ESCs and EpiLCs upon Δ9-THC exposure could be carried through PGCLCs differentiation. Metabolic characterization revealed that PGCLCs arising from exposed ESCs and EpiLCs showed increased mitochondrial respiration. Thus, in the absence of direct or indirect continuous exposure, Δ9-THC still has lasting consequences on the metabolome of embryonic germ cells. A recent study in *Drosophila* reported that nutrient stress induces oocyte metabolites remodelling that drives the onset of metabolic diseases in the progeny (*Hocaoglu et al., 2021*). This indicates that non-DNA-associated factors, such as germline metabolites, can act as factors of inheritance. Similarly, we show here that exposure to Δ9-THC remodels ESCs and EpiLCs metabolome and that a metabolic memory of this exposure is retained during PGCLCs differentiation (*Figure 8*). In addition to metabolic remodelling, we show that the PGCLCs transcriptome is also altered. In particular, despite proliferation and a higher number of cells, the number of DEGs that were downregulated in PGCLCs deriving from Δ9-THC-exposed ESCs and EpiLCs suggests a general degradation of PGCLCs' homeostasis. Functional annotation further indicated that these downregulated genes are related to structural cellular components, to the interaction with the extracellular environment and, specifically, to cell adhesion and junction. During the development of the central nervous system, perinatal Δ9-THC exposure has also been associated with alteration in cell adhesion, with an impact on neuronal interactions and morphology (*Gómez et al., 2007*; *Keimpema et al., 2011*; *Kittler et al., 2000*). Cell-cell adhesion is crucial in PGCs' formation both in cell culture systems (*Okamura et al., 2003*) as well as in vivo where it controls PGCs motility during their migration towards the developing somatic gonad (*Barton et al., 2016*). Our results thus show that exposure to Δ9-THC prior to specification affects embryonic germ cells' transcriptome and metabolome. This in turn could have adverse consequences on cell-cell adhesion with an impact on PGC normal development in vivo.

Despite epidemiological evidence of cannabis exposure in parents being associated with adverse effects in the offspring (*Smith et al., 2020*; *Szutorisz and Hurd, 2018*), the molecular mechanisms involved in the inheritance of exposure have not been extensively studied. In vitro models for germ cell development offer a unique opportunity for such studies (*Verdikt et al., 2022*) and our work in mouse PGCLCs identifies metabolites as relevant carriers of information across developmental stages. While in vitro gametogenesis, up to the reconstitution of fully functional spermatozoa and oocytes, is a reality in mouse (*Hikabe et al., 2016*; *Komeya et al., 2018*; *Luo and Yu, 2021*; *Yamashiro et al., 2018*), transposition to humans has been hindered by ethical and technical considerations (*Luo and Yu, 2021*). For instance, protocols for human PGCLCs have been developed (*Gell et al., 2020*) but proper induction to viable, fertile offspring cannot be verified (*Luo and Yu, 2021*). Nevertheless, our preliminary results in hESCs show that Δ9-THC exposure negatively impacts their global energy metabolism (*Figure 3—figure supplement 1*). Future studies will need to confirm whether such metabolic reprogramming is also carried over developmental stages in human models for the germline.

Together, our studies reveal a moderate but significant impact of Δ9-THC exposure on early embryonic processes. Our work also highlights the importance of the metabolic remodelling induced by Δ9-THC and its potential role as a driver of exposure memory through differentiation stages.

## Materials and methods
### Cell culture and PGCLCs model

Mouse ESCs containing the two fluorescent reporters Blimp1::mVenus and Stella::ECFP (BVSC cells) were described previously (*Ohinata et al., 2008*). The female BVSC clone H18 and male clone R8 were kindly provided by Mitinori Saitou. Cells were seeded on coated plates (Poly-L-ornithine [0.001%; A-004-C; Sigma-Aldrich] and laminin [300 ng/mL; L2020; Sigma-Aldrich]) in 2i+LIF culture medium

(N2B27 Media, CHIR99021 [30 µM; NC9785126; Thermo Fisher], PD0325901 [10 µM; NC9753132; Thermo Fisher], ESGRO Leukemia Inhibitory Factor (LIF) [1000 U/mL, ESG1106; Sigma-Aldrich]) for 48 hr. Differentiation of ESCs to EpiLCs was performed by seeding the cells on Human Plasma Fibronectin (HPF)-coated plates [16.7 µg/mL; 33016015; Thermo Fisher] in the presence of EpiLC induction medium (N2B27 medium containing activin A [20 ng/mL; 50-398-465; Thermo Fisher]), basic fibroblast growth factor (bFGF) [12 ng/mL; 3139FB025; R&D Systems], and KnockOut Serum Replacement [KSR, 1%; Thermo Fisher]. For PGCLCs induction, 44 hr EpiLCs were harvested using TrypLE Select (1 X) (Thermo Fisher) and seeded either in 96-wells plate (Nunclon Sphera, Thermo Fisher) or in EZsphere plates for large-scale induction (Nacalai) in the presence of GK15 medium (Glasgow's Minimal Essential Medium [GMEM, 11710035, Thermo Fisher]) supplemented with 15% KSR, 0.1 mM Minimal Essential Medium Nonessential Amino Acids [MEM-NEAA], 1 mM sodium pyruvate, 0.1 mM 2-mercaptoethanol, 100 U/mL penicillin, 0.1 mg/mL streptomycin, and 2 mM L-glutamine in the presence of bone morphogenetic protein 4 (BMP4; 500 ng/mL; 5020 BP-010/CF; R&D Systems), LIF, stem cell factor (SCF; 100 ng/mL; 50-399-595; R&D Systems), bone morphogenetic protein 8b (BMP8b; 500 ng/mL; 7540 BP-025; R&D Systems), and epidermal growth factor (EGF; 50 ng/mL; 2028EG200; R&D Systems). Cells were cultured for 5d before collection, dissociation of embryoid bodies and downstream experiments.

The hESCs UCLA2 cells (*Diaz Perez et al., 2012*) were cultured on plates coated with Recombinant laminin-511 E8 (iMatrix-511 Silk, 892 021, Amsbio) and were maintained under a feeder-free condition in the StemFit Basic03 medium (SFB-503, Ajinomoto) containing bFGF (100-18B, Peprotech). Prior to passaging, hESC cultures were treated with a 1:1 mixture of TrypLE Select (12563011, Thermo Fisher) and 0.5 mM EDTA/PBS for 15 min at 37 °C to dissociate them into single cells. For routine maintenance, hESCs were plated into a six-well plate (3516, Corning) at a density of $2 \times 10^3$ cells/cm2 with 10 µM ROCK inhibitor (Y-27632; Tocris, 1254) added in culture medium for 1 day after hESCs passaging. hESCs were plated into 24-wells plate (3526, Corning) at 10,000 cells per well for the viability and viable cell count. For WST-1 assays, hESCs were seeded in 96-wells plate (353072, Falcon) at 1500 cells per well.

All cells were cultured in a humidified environment at 37 °C under 5% $CO_2$. All cells were tested negative for mycoplasma by a PCR test (ATCC ISO 9001:2008 and ISO/IEC 17025:2005 quality standards).

## Δ9-THC exposures

To assess the impact of Δ9-THC exposure on the developmental trajectory of PGCLCs, three exposure schemes were tested: (1) ESCs exposure only, (2) EpiLCs exposure only, and (3) ESCs +EpiLCs exposure. The stock of Δ9-THC was obtained from the National Institute on Drug Abuse (7370–023 NIDA; Bethesda, MD). The stock was adjusted to a concentration of 200 mM diluted in ethanol, aliquoted and stored according to the DEA's recommendations. The dose range of 0–100 µM was determined based on Δ9-THC physiological measurements in the blood, plasma, and follicular fluid (*Fuchs Weizman et al., 2021*; *Hunault et al., 2009*; *Pacifici et al., 2020*). For each exposure, new aliquots of Δ9-THC were diluted in ESCs or EpiLCs culture media in coated tubes (Sigmacote, Sigma Aldrich). Exposure was performed for 48 hr. Solubility tests were performed and ethanol was added to reach the same amount for each Δ9-THC concentration (0.05% ethanol). Vehicle control corresponded to 0.05% ethanol added to the respective culture media for ESCs or EpiLCs. All experiments performed are authorized under DEA registration number RA0546828.

## Δ9-THC quantification

Samples were treated with 500 µl of 1% Formic Acid (543804, Sigma Aldrich). Labelled Tetrahydrocannabinol (THC-d3), used as an internal standard, was added to every sample to account for compound loss during sample processing. Samples were then mixed vigorously and centrifuged at 16,000 *g* for 5 min at room temperature. The supernatants were loaded into phospholipid removal cartridges (Phenomenex Phree) and the eluents were dried down in a vacuum concentrator. Samples were reconstituted in 30 µl of HPLC-grade water, vortexed rigorously, and centrifuged at 16,000 *g* for 5 min at room temperature. The supernatant was transferred to HPLC vials and 15 µl were injected for analysis onto a hybrid linear ion trap/orbitrap mass spectrometer (Thermo Scientific LTQ Orbitrap XL, UCLA Pasarow Mass Spectrometry Lab). For specificity and accurate quantitative measurement, the mass

spectrometer was set to fragment preselected precursor ions for THC and THC-d3 under standard MS/MS fragmentation conditions in positive ion mode. The mass spectrometer was coupled to a Dionex Ultimate 300 HPLC (Thermo Scientific) with a reversed phase Phenomenex analytical column (Kinetex 1.7 µm Polar C18 100 Å 100x2.1 mm) equilibrated in eluant A (water/formic acid, 100/0.1, v/v) and eluted (100 µl/min) with a linearly increasing concentration of eluant B (acetonitrile/formic acid, 100/0.1, v/v; min/%B, 0/5, 5/5, 8/95, 13/95, 14/5, 25/5). Data was collected and processed with instrument manufacturer-supplied software Xcalibur 2.07. A set of standard curve samples were prepared in cell culture media for each experiment. Samples and standards were prepared in duplicates. The standard curve was made by plotting the known concentration of THC per standard against the ratio of measured chromatographic peak areas corresponding to the THC over that of the IS THC-d3 (analyte/IS). The trendline equation was then utilized to calculate the absolute concentrations of the THC in cell culture.

## PGCLCs induction efficiency

Changes in PGCLCs induction were calculated by flow cytometry. Practically, d5 aggregates were harvested, dissociated using TrypLE Select, and resuspended in fluorescence-activated cell sorting (FACS) buffer (1×Dulbecco's phosphate buffered saline [DPBS], 1% BSA, 1 mM EDTA, 25 mM HEPES). Quantification of subfractions of double-positive PGCLCs (Blimp1::mVenus +and Stella::ECFP+), single-positive (Blimp1::mVenus+) and double-negative cells was performed on a BD Biosciences LSRII (UCLA BSCRC Flow Cytometry Core). Cells were initially identified by forward- and side-scatter gating, with back-gating used to verify the accuracy by which target cell populations were identified. Cell populations of interest were identified by 2-D plots displaying the parameter of interest, using embryoid bodies cultured in GK15 medium without added cytokines and BMPs as a negative control. Fluorescent compensation beads were used as positive controls and to calculate the spectral overlap (Thermo Fisher, A10514 and 01-2222-42 adsorbed to a CD45 Pacific Blue antibody [OB180026], serving as compensation control for mVenus and ECFP, respectively). Manually defined gates as well as quadrants were used, as indicated. The FlowJo software was used to calculate the percentage of induction and generate the associated graphs (version 10, FlowJo, LLC).

## Cell viability and proliferation studies

The viability and viable cell count of ESCs and EpiLCs were calculated using Trypan blue (0.4%, Thermo Fisher) on a Countess II FL Automated Cell Counter (Thermo Fisher). For BrdU incorporation studies, cells were permeabilized, fixed, and stained using the BrdU Flow Kit (PerCP-Cy5.5 Mouse anti-BrdU, BD Biosciences) before analysis by flow cytometry on a BD Biosciences LSRII (UCLA BSCRC Flow Cytometry Core). Quantification of PGCLCs proliferation was performed using CellTrace Yellow (5 µM, added at the induction, Thermo Fisher), which binds to intracellular amines after diffusing through cell membranes. The overall fluorescent signal, which gradually decreases as cell division occurs, reflects the number of cell divisions occurring and was measured on a BD Biosciences LSRII (UCLA BSCRC Flow Cytometry Core). The FlowJo software was used to calculate the percentage of induction, the number of cell divisions and generate the associated graphs (version 10, FlowJo, LLC).

## CB1 antagonist treatment

To block the effects of Δ9-THC on the cannabinoid receptor CB1, ESCs were plated on 48-well plate and were pre-treated with 1 µM of SR141716/Rimonabant (SML0800, Sigma Aldrich) for 1 hr before being exposed to the dose range of Δ9-THC, as above. After 24 hr incubation, this procedure was repeated and cells were harvested after 48 hr total incubation. The viability and viable cell count was calculated using Trypan blue (0.4%, Thermo Fisher) on a Countess II FL Automated Cell Counter (Thermo Fisher). The concentration of 1 µM of Rimonabant was chosen based on previous experiments (*Lojpur et al., 2019*) and did not impact cell viability nor cell number on its own.

## Western blotting

Membrane proteins were extracted from cell pellets using the Mem-PER Plus Membrane Protein Extraction Kit (89842, Thermo Fisher) according to the manufacturer's protocol. Western blotting was performed with 25 µg of protein extracts. The immunodetection was assessed using primary antibodies targeting CB1 (101500, Cayman Chemical, RRID: AB_327840) or β-actin (3700, Cell Signaling

Technology, RRID:AB_2242334) as loading control. Horseradish peroxidase (HRP)-conjugated secondary antibodies were used for chemiluminescence detection (Amersham).

## WST-1 assay

The colorimetric assay WST-1 was used according to the manufacturer's instructions (Roche). The tetrazolium salt WST-1 is reduced by mitochondrial dehydrogenases to formazan using NAD(P)H as co-substrates. Thus, the quantity of formazan is directly proportional to NAD(P)$^+$.

## Mitochondrial activity

Staining for mitochondria was performed by incubating cells at 37 °C with 250 nM MitoTracker CMXRos (M7512, Thermo Fisher) for 30 min (*Pendergrass et al., 2004*). Cells were washed and analyzed by flow cytometry on a BD Biosciences LSRII (UCLA BSCRC Flow Cytometry Core). The FlowJo software (version 10, FlowJo, LLC) was used to calculate the mean fluorescence intensity (MFI) corresponding to the average fluorescence intensity of each event of the selected cell population within the chosen fluorescence channel associated to MitoTracker CMXRos.

## Seahorse experiments

The extracellular acidification rate (ECAR) and the oxygen consumption rate (OCR) are indicative of glycolysis and mitochondrial respiration, respectively. A total of $10 \times 10^3$ ESCs and $8 \times 10^3$ EpiLCs were seeded on Seahorse XF96 plates (101085–004, Agilent Technologies) and exposed to increasing doses of Δ9-THC for 48 hr. On the day of the assay, cells were washed with assay medium (unbuffered DMEM assay medium [5030, Sigma Aldrich] supplemented with 31.6 mM NaCl, 3 mg/L phenol red, 5 mM HEPES, 5 mM glucose, 2 mM glutamine and 1 mM sodium pyruvate). For OCR measurement, compounds were injected sequentially during the assay resulting in final concentrations of 2 µM oligomycin, 0.75 µM and 1.35 µM FCCP, 1 µM rotenone and 2 µM antimycin. ECAR was measured in parallel. The measured quantities were normalized to the protein content as measured by a BCA quantitation (23227, Thermo Fisher).

## Mass spectrometry-based metabolomics analysis

To extract intracellular metabolites, cells were rinsed with cold 150 mM ammonium acetate (pH 7.3) then incubated with 80% ice-cold methanol supplemented with 10 nmol D/L-norvaline for 1 hr. Following resuspension, cells were pelleted by centrifugation (15,000 *g*, 4 °C for 15 min). The supernatant was transferred into a glass vial and metabolites were dried down under vacuum, then resuspended in 70% acetonitrile. Mass spectrometry analysis was performed at the UCLA Metabolomics Center with an UltiMate 3000RSLC (Thermo Scientific) coupled to a Q Exactive mass spectrometer (Thermo Scientific) in polarity-switching mode with positive voltage 3.0 kV and negative voltage 2.25 kV. Separation was achieved using a gradient elution with (A) 5 mM NH4AcO (pH 9.9) and (B) acetonitrile. The gradient ran from 15% (A) to 90% (A) over 18 min, followed by an isocratic step for 9 min and re-equilibration for 7 min. Metabolites were quantified as area under the curve based on retention times and using accurate mass measurements (≤3 ppm) with the TraceFinder 3.1 software (Thermo Scientific). For heatmap depiction, the relative amounts of metabolites were normalized to the mean value across all samples for the same condition and to the number of viable cells harvested in parallel on a control plate. Pathway enrichment for up- and downregulated KEGG metabolites (|log2(fold-change)|=0.25) was determined using the MetaboAnalyst 5.0 platform (https://www.metaboanalyst.ca)(*Pang et al., 2022*).

## RNA-sequencing

Total RNA was extracted from ESCs and EpiLCs pellets using the AllPrep DNA/RNA Micro Kit (Qiagen), according to the manufacturer's protocol. For PGCLCs, d5 embryoid bodies were harvested and cells were dissociated using TrypLE Select followed by resuspension in fluorescence-activated cell sorting (FACS) buffer (1×Dulbecco's phosphate buffered saline [DPBS], 1% BSA, 1 mM EDTA, 25 mM HEPES) and cell suspension were passed through a cell strainer (70 µm). Cells were sorted on a BD Biosciences FACSAria III (UCLA BSCRC Flow Cytometry Core). Practically, cell populations of interest, being double-positive (Blimp1::mVenus +and Stella::ECFP+) were sorted and collected in microtubes containing GK15 medium. Total RNA was extracted from double-positive PGCLCs using the AllPrep

DNA/RNA Micro Kit (Qiagen). RNA concentration was measured using a NanoDrop 2000 UV spectrophotometer (Thermo Fisher). Libraries were prepared with the KAPA mRNA HyperPrep Kit (BioMek) or with the RNA library prep kit (ABClonal) following the manufacturers' protocols. Briefly, poly(A) RNA were selected, fragmented and double-stranded cDNA synthesized using a mixture of random and oligo(dT) priming, followed by end repair to generate blunt ends, adaptor ligation, strand selection, and polymerase chain reaction (PCR) amplification to produce the final library. Different index adaptors were used for multiplexing samples in one sequencing lane. Sequencing was performed on an Illumina NovaSeq 6000 sequencers for paired end (PE), 2×150 base pair (bp) runs. Data quality check was performed using Illumina Sequencing Analysis Viewer (SAV) software. Demultiplexing was performed with Illumina Bcl2fastq2 program (version 2.19.1.403; Illumina Inc).

## Differential gene expression analysis

The quality of the reads was verified using FastQC (*Andrews, 2010*) before reads were aligned to the mm10 reference genome (GRCm39) using STAR (*Dobin et al., 2013*) with the following arguments: `--readFilesCommand` zcat `--outSAMtype` BAM SortedByCoordinate `--quantMode` GeneCounts `--outFilterMismatchNmax` 5 `--outFilterMultimapNmax` 1. The quality of the resulting alignments was assessed using QualiMap (*García-Alcalde et al., 2012*). The Python package HTseq was used for gene counts (*Anders et al., 2015*) using the following arguments: `--stranded=no --idattr=gene_id --type=exon --mode=union` r pos `--format=bam`. Output files were filtered to remove genes with low count (≤10) then were used for differential gene expression analysis using DESeq2 (*Love et al., 2014*). The negative binomial regression model of ComBat-seq was used to correct unwanted batch effects (*Zhang et al., 2020*). For a gene to be classified as showing differential expression between treated and untreated cells, a threshold of |log2(fold-change)|=0.5 and Benjamini-Hochberg adjusted p-value ≤0.05 had to be met.

## Gene ontology (GO) analysis

Lists of differentially expressed genes were generated from read counts using DESeq2 Bioconductor package (*Love et al., 2014*). Enrichment of GO terms in lists of up- and downregulated genes (|log2(fold-change)|=0.25) was determined using g:Profiler (*Raudvere et al., 2019*). Redundant GO terms were removed using reduce + visualize gene ontology (REVIGO) (*Supek et al., 2011*). Terms were included if the fold enrichment (frequency of DEGs in each GO term to the frequency of total genes in GO terms) was higher than 1.5 and if the Benjamini-Hochberg-adjusted p-value was less than 0.05. Plots for GO terms were generated using an R script (*Bonnot et al., 2019*).

## Statistical methods

Statistical analyses were performed using GraphPad Prism 9 software. Unless otherwise specified, two-tailed T-tests were performed on pairwise comparisons for testing of significance. In all cases, significance was determined by p-values less than or equal to 0.05. Each figure corresponds to at least three independent biological repeats with three technical replicates (N=3, n=3), unless otherwise specified. Number of asterisks on plots indicates the level of statistical significance: *(p<0.05), **(p<0.01), ***(p<0.001), ****(p<0.0001).

## Acknowledgements

We thank Jessica Scholes, Felicia Codrea and Jeffrey Calimlim from the UCLA BSCRC Flow Cytometry Core for their assistance in cell sorting. We thank Linsey Stiles (UCLA Mitochondria and Metabolism Core), Johanna ten Hoeve-Scott and Thomas Graeber (UCLA Metabolomics Center) for their technical support in the metabolomics analyses. We thank Xinmin Li and his team at the UCLA Technology Center for Genomics & Bioinformatics for their technical support in high-throughput sequencing. We thank Gazmend Elezi and Julian Whitelegge (UCLA Pasarow Mass Spectrometry Laboratory) for their help in Δ9-THC quantification. We thank Luigi Bellocchio and Giovanni Marsicano for their recommendations in Δ9-THC exposure setups. We would like to acknowledge the UCLA Cannabis Research Initiative (UCRI) and Dr Ziva Cooper for their continuous support and encouragement. PA is supported by NIEHS R01 ES027487, the John Templeton Foundation Grant 60742, and the Iris Cantor-UCLA Women's Health Center and the NCATS UCLA CTSI Grant Number UL1TR001881. RV is

a postdoctoral fellow and acknowledges support from the Belgian-American Educational Foundation (BAEF). ATC is supported by NIH/NICHD R01HD079546 (ATC).

## Additional information

### Funding

| Funder | Grant reference number | Author |
|---|---|---|
| National Institute for Environmental Health Sciences | R01 ES027487 | Patrick Allard |
| John Templeton Foundation | 60742 | Patrick Allard |
| Eunice Kennedy Shriver National Institute of Child Health and Human Development | R01HD079546 | Amander T Clark |
| Iris Cantor-UCLA Women's Health Center and the NCATS UCLA CTSI | UL1TR001881 | Roxane Verdikt |

The funders had no role in study design, data collection and interpretation, or the decision to submit the work for publication.

### Author contributions

Roxane Verdikt, Conceptualization, Data curation, Formal analysis, Validation, Investigation, Visualization, Methodology, Writing - original draft, Project administration, Writing - review and editing; Abigail A Armstrong, Data curation, Formal analysis, Investigation, Methodology; Jenny Cheng, Data curation, Formal analysis, Methodology; Young Sun Hwang, Resources, Methodology; Amander T Clark, Resources, Supervision, Writing - review and editing; Xia Yang, Resources, Supervision, Visualization, Writing - original draft, Writing - review and editing; Patrick Allard, Conceptualization, Resources, Supervision, Funding acquisition, Investigation, Writing - original draft, Project administration, Writing - review and editing

### Author ORCIDs

Roxane Verdikt http://orcid.org/0000-0002-1091-6720
Xia Yang http://orcid.org/0000-0002-3971-038X
Patrick Allard http://orcid.org/0000-0001-7765-1547

Reviewer #1 (Public Review): https://doi.org/10.7554/eLife.88795.3.sa1
Reviewer #2 (Public Review): https://doi.org/10.7554/eLife.88795.3.sa2
Author Response https://doi.org/10.7554/eLife.88795.3.sa3

## Additional files

### Supplementary files

• MDAR checklist

• Supplementary file 1. Expression of epigenetic enzymes in Δ9-THC or control exposed ESCs and EpiLCs as measured by RNA-seq.

### Data availability

The RNA sequencing data from this study is made available at the Gene Expression Omnibus (GEO) under the following accession number GSE226955.

The following dataset was generated:

| Author(s) | Year | Dataset title | Dataset URL | Database and Identifier |
|---|---|---|---|---|
| Verdikt R, Armstrong AA, Cheng J, Yang X, Allard P | 2023 | Adverse effects of gestational Δ9-tetrahydrocannabinol exposure on the embryonic germline transcriptome and metabolome | https://www.ncbi.nlm.nih.gov/geo/query/acc.cgi?acc=GSE226955 | NCBI Gene Expression Omnibus, GSE226955 |

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
