## [Editor Report · eLife assessment]

The study presents **valuable** findings demonstrating that physiologically relevant concentrations delta-9-tetrahydrocannabinol, which is found in cannabis, have metabolic effects on early mouse embryonic cell types. The evidence supporting the claims is **convincing**. The work will be of interest to researchers in stem cell and epigenetics fields.

---

## [Referee Report · Reviewer #1 (Public Review)]

The study is valuable because it shows that physiologically relevant ∆9-THC concentrations have metabolic effects on early mouse embryonic cell types, which could cause developmental effects. Overall, the authors have convincing evidence showing that ∆9-THC has metabolic effects on mouse embryonic stem cells (mESCs), and that these effects persist in mESC-derived primordial germ cell-like cells even after ∆9-THC treatment has stopped. In this revised version, the authors included additional data to characterize the dose-dependence of the effects of ∆9-THC. Furthermore, they supported their finding of metabolic memory in PGCLCs by ruling out the potential alternative explanation that ∆9-THC persists in the cultured cells over the course of their experiment. This study has two significant implications: first, that ∆9-THC may alter the metabolism of early mouse embryos, and second, that mouse primordial germ cell-like cells can have a memory of previous metabolic perturbations.

The authors investigated the metabolic effects of ∆9-THC, the main psychoactive component of cannabis, on early mouse embryonic cell types. They found that ∆9-THC increases proliferation in male and female mouse embryonic stem cells (mESCs) and upregulates glycolysis. Additionally, primordial germ cell-like cells (PGCLCs) differentiated from ∆9-THC-exposed cells also show alterations to their metabolism. The study is valuable because it shows that physiologically relevant ∆9-THC concentrations have metabolic effects on cell types from the early embryo, which may cause developmental effects. Intriguingly, these effects persist in PGCLCs even after withdrawal of ∆9-THC.

The study shows that ∆9-THC increases the proliferation rate of mESCs but not mEpiLCs, without substantially affecting cell viability, except at the highest dose of 100 µM which shows toxicity (Figure 1). The dose required to cause increased proliferation was approximately 1 nM (Supplementary Figure 1), which is remarkably low. Treatment of mESCs with rimonabant (a CB1 receptor antagonist) blocks the effect of 100 nM ∆9-THC on cell proliferation, showing that the proliferative effect is mediated by CB1 receptor signaling. Similarly, treatment with 2-deoxyglucose, a glycolysis inhibitor, also blocks this proliferative effect (Figure 4G-H). Therefore, the effect of ∆9-THC depends on both CB1 signaling and glycolysis. This set of experiments strengthens the conclusions of the study by helping to elucidate the mechanism of the effects of ∆9-THC.

The study also profiles the transcriptome and metabolome of cells exposed to 100 nM ∆9-THC (Figure 4). Although the transcriptomic changes are modest overall, there is upregulation of anabolic genes, consistent with the increased proliferation rate in mESCs. Metabolomic profiling revealed a broad upregulation of metabolites in mESCs treated with 100 nM ∆9-THC. Some metabolic effects were also observed at a lower dose of 10 nM (Figure 3B).

Additionally, the study shows that ∆9-THC can influence germ cell specification. mESCs were differentiated to mEpiLCs in the presence or absence of ∆9-THC, and the mEpiLCs were subsequently differentiated to mPGCLCs. mPGCLC induction efficiency was tracked using a BV:SC dual fluorescent reporter. ∆9-THC treated cells had a moderate increase in the double positive mPGCLC population, and decrease in the double negative population. A cell tracking dye showed that mPGCLCs differentiated from ∆9-THC treated cells had undergone more divisions on average. As with the mESCs, these mPGCLCs also had altered gene expression and metabolism, consistent with an increased proliferation rate. Importantly, in the revised version, the authors supported their finding of metabolic memory in mPGCLCs by ruling out the potential alternative explanation that ∆9-THC persists in the cultured cells over the course of their experiment (Supplementary Figure 13).

Finally, the authors also observed that ∆9-THC decreases the proliferation of human ESCs (Supplementary Figure 4). These cells were in the primed pluripotent state, making them more similar to mEpiLCs than mESCs. Although this result may form the basis of follow-up experiments in a separate paper, it would be premature to conclude that the same effects will be observed in human cells as were observed in mouse cells.

Overall, this study provides good evidence for ∆9-THC having metabolic effects on mouse ESCs, and additionally shows that these effects can persist during germ cell specification. Potential effects of ∆9-THC exposure during early embryonic development are important for society to understand, and the results of this study are significant for public health.

---

## [Referee Report · Reviewer #2 (Public Review)]

Verdikt et al. focused on the influence of Δ9-THC, the most abundant phytocannabinoid, on early embryonic processes. The authors chose an in vitro differentiation system as model, and compared the proliferation rate, metabolic status and transcriptional level in ESCs, exposure to Δ9-THC. They also evaluated the change of metabolism and transcriptome in PGCLCs derived from Δ9-THC-exposed cells. All the methods in this paper do not involve the differentiation of ESCs to lineage specific cells. So the results cannot demonstrate the impact of Δ9-THC on postimplantation developmental stages. In brief, the authors want to explore the impact of Δ9-THC on preimplantation developmental stages, but they only detected the change in ESCs and PGCLCs derived from ESCs, exposure to Δ9-THC, which showed the molecular characterization of the impact of Δ9-THC exposure on ESCs and PGCLCs.

---

## [Author Response]

The following is the authors’ response to the original reviews.

We want to thank the Editor and Reviewers for their thorough assessment of the manuscript as well as their constructive critiques. We have collated below the public review and recommendations from each Reviewer as well as our responses to them.

**eLife assessment**
This study by Verdikt et al. provided solid evidence demonstrating the potential impacts of Δ9-tetrahydrocannabinol (Δ9-THC) on early embryonic development using mouse embryonic stem cells (mESCs) and in vitro differentiation. Their results revealed that Δ9-THC enhanced mESCs proliferation and metabolic adaptation, possibly persisting through differentiation to Primordial Germ Cell-Like Cells (PGCLCs), though the evidence supporting this persistence was incomplete. Although the study is important, it was limited by being conducted solely in vitro and lacking parallel human model experiments.
**Reviewer #1 (Public Review):**
The authors investigated the metabolic effects of ∆9-THC, the main psychoactive component of cannabis, on early mouse embryonic cell types. They found that ∆9-THC increases proliferation in female mouse embryonic stem cells (mESCs) and upregulates glycolysis. Additionally, primordial germ cell-like cells (PGCLCs) differentiated from ∆9-THC-exposed cells also show alterations to their metabolism. The study is valuable because it shows that physiologically relevant ∆9-THC concentrations have metabolic effects on cell types from the early embryo, which may cause developmental effects. However, the claim of "metabolic memory" is not justified by the current data, since the effects on PGCLCs could potentially be due to ∆9-THC persisting in the cultured cells over the course of the experiment, even after the growth medium without ∆9-THC was added.The study shows that ∆9-THC increases the proliferation rate of mESCs but not mEpiLCs, without substantially affecting cell viability, except at the highest dose of 100 µM which shows toxicity (Figure 1). Treatment of mESCs with rimonabant (a CB1 receptor antagonist) blocks the effect of 100 nM ∆9-THC on cell proliferation, showing that the proliferative effect is mediated by CB1 receptor signaling. Similarly, treatment with 2-deoxyglucose, a glycolysis inhibitor, also blocks this proliferative effect (Figure 4G-H). Therefore, the effect of ∆9-THC depends on both CB1 signaling and glycolysis. This set of experiments strengthens the conclusions of the study by helping to elucidate the mechanism of the effects of ∆9-THC.Although several experiments independently showed a metabolic effect of ∆9-THC treatment, this effect was not dose-dependent over the range of concentrations tested (10 nM and above). Given that metabolic effects were observed even at 10 nM ∆9-THC (see for example Figure 1C and 3B), the authors should test lower concentrations to determine the dose-dependence and EC50 of this effect. The authors should also compare their observed EC50 with the binding affinity of ∆9-THC to cellular receptors such as CB1, CB2, and GPR55 (reported by other studies).The study also profiles the transcriptome and metabolome of cells exposed to 100 nM ∆9-THC. Although the transcriptomic changes are modest overall, there is upregulation of anabolic genes, consistent with the increased proliferation rate in mESCs. Metabolomic profiling revealed a broad upregulation of metabolites in mESCs treated with 100 nM ∆9-THC.Additionally, the study shows that ∆9-THC can influence germ cell specification. mESCs were differentiated to mEpiLCs in the presence or absence of ∆9-THC, and the mEpiLCs were subsequently differentiated to mPGCLCs. mPGCLC induction efficiency was tracked using a BV:SC dual fluorescent reporter. ∆9-THC treated cells had a moderate increase in the double positive mPGCLC population and a decrease in the double negative population. A cell tracking dye showed that mPGCLCs differentiated from ∆9-THC treated cells had undergone more divisions on average. As with the mESCs, these mPGCLCs also had altered gene expression and metabolism, consistent with an increased proliferation rate.My main criticism is that the current experimental setup does not distinguish between "metabolic memory" vs. carryover of THC (or its metabolites) causing metabolic effects. The authors assume that their PGCLC induction was performed "in the absence of continuous exposure" but this assumption may not be justified. ∆9-THC might persist in the cells since it is highly hydrophobic. In order to rule out the persistence of ∆9-THC as an explanation of the effects seen in PGCLCs, the authors should measure concentrations of ∆9-THC and THC metabolites over time during the course of their PGCLC induction experiment. This could be done by mass spectrometry. This is particularly important because 10 nM of ∆9-THC was shown to have metabolic effects (Figure 1C, 3B, etc.). Since the EpiLCs were treated with 100 nM, if even 10% of the ∆9-THC remained, this could account for the metabolic effects. If the authors want to prove "metabolic memory", they need to show that the concentration of ∆9-THC is below the minimum dose required for metabolic effects.Overall, this study is promising but needs some additional work in order to justify its conclusions. The developmental effects of ∆9-THC exposure are important for society to understand, and the results of this study are significant for public health.**Reviewer #1 (Recommendations For The Authors):*This has the potential to be a good study, but it's currently missing two key experiments:What is the minimum dose of ∆9-THC required to see metabolic effects?

We would like to thank Reviewer 1 for their insightful comments. We have included exposures to lower doses of ∆9-THC in Supplementary Figure 1. Our data shows that ∆9-THC induces mESCs proliferation from 1nM onwards. However, when ESCs and EpiLCs were exposed to 1nM of ∆9-THC, no significant change in mPGCLCs induction was observed (updated Figure 6B). Of note, in their public review, Reviewer 1 mentioned that “The authors should also compare their observed EC50 with the binding affinity of ∆9-THC to cellular receptors such as CB1, CB2, and GPR55 (reported by other studies).” According to the literature, stimulation of non-cannabinoid receptors and ion channels (including GPR18, GPR55, TRPVs, etc.) occurs at 40nM-10µM of ∆9-THC (Banister et al., 2019). We therefore expect that at the lower nanomolar range tested, CB1 is the main receptor stimulated by ∆9-THC, as we showed for the 100nM dose in our rimonabant experiments (Fig. 2).

Is the residual THC concentration during the PGCLC induction below this minimum dose? Even if the effects are due to residual ∆9-THC, this would not undermine the overall study. There would simply be a different interpretation of the results.

This experiment was particularly important to distinguish between a “true” ∆9-THC metabolic memory or residual ∆9-THC leftover during PGCLCs differentiation. Our mass spectrometry quantification revealed that no significant ∆9-THC could be detected in day 5 embryoid bodies compared to treated EpiLCs prior to differentiation (Supplementary Figure 13). These results support the existence of ∆9-THC metabolic memory across differentiation.

You also do not mention whether you tested your cells for mycoplasma. This is important since mycoplasma contamination is a common problem that can cause artifactual results. Please test your cells and report the results.

All cells were tested negative for mycoplasma by a PCR test (ATCC ISO 9001:2008 and ISO/IEC 17025:2005 quality standards). This information has been added in the Material and Methods section.

Minor points:1. I don't think it's correct to say that cannabis is the most commonly used psychoactive drug. Alcohol and nicotine are more commonly used. See: https://nida.nih.gov/research-topics/alcohol and https://www.cancer.gov/publications/dictionaries/cancer-terms/def/psychoactive-substance I looked at the UN drugs report [ref 1] and alcohol or nicotine were not included on that list of drugs, so the UN may use a different definition. This doesn't affect the importance or conclusions of this study, but the wording should be changed.

We agree and are now following the WHO description of cannabis (https://www.who.int/teams/mental-health-and-substance-use/alcohol-drugs-and-addictive-behaviours/drugs-psychoactive/cannabis) by referring to it as the “most widely used illicit drug in the world”. (Line 44).

1. It would be informative to use your RNA-seq data to examine the expression of receptors for ∆9-THC such as CB1, CB2, and GPR55. CB1 might be the main one, but I am curious to see if others are present.

We have explored the protein expression of several cannabinoid receptors, including CB2, GPR18, GPR55 and TRPV1 (Bannister et al., 2019). These proteins, except TRPV1, were lowly expressed in mouse embryonic stem cells compared to the positive control (mouse brain extract, see Author response image 1). Furthermore, our experiment with Rimonabant showed that the proliferative effects of ∆9-THC are mediated through CB1.

**Author response image 1. sa3fig1:** Cannabinoid receptors and non-cannabinoid receptors protein expression in mouse embryonic stem cells.

1. Make sure to report exact p-values. You usually do this, but there are a few places where it says p<0.0001. Also, report whether T-tests assumed equal variance (Student's) or unequal variance (Welch's). [In general, it's better to use unequal variance, unless there is good reason to assume equal variance.]

Prism, which was used for statistical analyses, only reports p-values to four decimal places. For all p-values that were p<0.0001, the exact decimals were calculated in Excel using the “=T.DIST.2T(t, df)” function, where the Student’s distribution and the number of degrees of freedom computed by Prism were inputted. Homoscedasticity was confirmed for all statistical analyses in Prism.

1. Figure 2A: An uncropped gel image should be provided as supplementary data. Additionally, show positive and negative controls (from cells known to either express CB1 or not express CB1)

The uncropped gel image is presented in Author response image 2. The antibody was validated on mouse brain extracts as a positive control as shown in Figure 1.

**Author response image 2. sa3fig2:** Uncropped gel corresponding to Fig. 2A where an anti-CB1 antibody was used.

1. Figure 6B: Please show a representative gating scheme for flow cytometry (including controls) as supplementary data. Also, was a live/dead stain used? What controls were used for compensation? These details should be reported.

The gating strategy is presented in Supplementary Figure 11. The Material and Methods section has also been expanded.

1. As far as I can tell, you only used female mESCs. It would be good to test the effects on male mESCs as well since these have some differences due to differences in X-linked gene expression (female mESCs have two active X chromosomes). I understand that you might not have a male BV:SC reporter line, so it would be acceptable to omit the mPGCLC experiments on male cells.

We have tested the 10nM-100µM dose range in the male R8 mESCs (Supplementary Figure 3). Similar results as with the female H18 cells were observed. Accordingly, PGCLCs induction was increased when R8 ESCs + EpiLCs were exposed to 100nM of ∆9-THC (Supplementary Figure 12). This is in line with ∆9-THC impact on fundamentally conserved metabolic pathways across species and sex, although it should be noted that one representative model of each sex is not sufficient to exclude sex-specific effects.

**Reviewer #2 (Public Review):**
In the study conducted by Verdikt et al, the authors employed mouse Embryonic Stem Cells (ESCs) and in vitro differentiation techniques to demonstrate that exposure to cannabis, specifically Δ9-tetrahydrocannabinol (Δ9-THC), could potentially influence early embryonic development. Δ9-THC was found to augment the proliferation of naïve mouse ESCs, but not formative Epiblast-like Cells (EpiLCs). This enhanced proliferation relies on binding to the CB1 receptor. Moreover, Δ9-THC exposure was noted to boost glycolytic rates and anabolic capabilities in mESCs. The metabolic adaptations brought on by Δ9-THC exposure persisted during differentiation into Primordial Germ Cell-Like Cells (PGCLCs), even when direct exposure ceased, and correlated with a shift in their transcriptional profile. This study provides the first comprehensive molecular assessment of the effects of Δ9-THC exposure on mouse ESCs and their early derivatives. The manuscript underscores the potential ramifications of cannabis exposure on early embryonic development and pluripotent stem cells. However, it is important to note the limitations of this study: firstly, all experiments were conducted in vitro, and secondly, the study lacks analogous experiments in human models.
**Reviewer #2 (Recommendations For The Authors):**
1. EpiLCs, characterized as formative pluripotent stem cells rather than primed ones, are a transient population during ESC differentiation. The authors should consider using EpiSCs and/or formative-like PSCs (Yu et al., Cell Stem Cell, 2021; Kinoshita et al., Cell Stem Cell, 2021), and amend their references to EpiLCs as "formative".

Indeed, EpiLCs are a transient pluripotent stem cell population that is “functionally distinct from both naïve ESCs and EpiSCs” and “enriched in formative phase cells related to pre-streak epiblast” (Kinoshita et al., Cell Stem Cell, 2021). Here, we used the differentiation system developed by M. Saitou and colleagues to derive PGCLCs (Hayashi et al, 2011). Since EpiSCs are refractory to PGCLCs induction (Hayashi et al, 2011), we used the germline-competent EpiLCs and took advantage of a well-established differentiation system to derive mouse PGCLCs. Most authors, however, agree that in terms of epigenetic and metabolic profiles, mouse EpiLCs represent a primed pluripotent state. We have added that PGCs arise in vivo “from formative pluripotent cells in the epiblast” on lines 85-86.

1. Does the administration of Δ9-THC, at concentrations from 10nM to 1uM, alter the cell cycle profiles of ESCs?

The proliferation of ESCs was associated with changes in the cell cycle, as presented in the new Supplementary Figure 2, which we discuss in lines 118-123.

1. Could Δ9-THC treatment influence the differentiation dynamics from ESCs to EpiLCs?

No significant changes were observed in the pluripotency markers associated with ESCs and EpiLCs (Supplementary Figure 9). We have added this information in lines 277-279.

1. The authors should consider developing knockout models of cannabinoid receptors in ESCs and EpiLCs (or EpiSCs and formative-like PSCs) for control purposes.

This is an excellent suggestion. Due to time and resource constraints, however, we focused our mechanistic investigation of the role of CB1 on the use of rimonabant which revealed a reversal of Δ9-THC-induced proliferation at 100nM.

1. Lines 134-136: "Importantly, SR141716 pre-treatment, while not affecting cell viability, led to a reduced cell count compared to the control, indicating a fundamental role for CB1 in promoting proliferation." Regarding Figure 2D, does the Rimonabant "+" in the "mock" group represent treatment with Rimonabant only? If that's the case, there appears to be no difference from the Rimonabant "-" mock. The authors should present results for Rimonabant-only treatment.

To be able to compare the effects +/- Rimonabant and as stated in the figure legend, each condition was normalized to its own control (mock with, or without Rimonabant). Author response image 3 is the unnormalized data showing the same effects of Δ9-THC and Rimonabant on cell number.

**Author response image 3. sa3fig3:** Unnormalized data corresponding to the Figure 2D.

1. In Figure 3, both ESCs and EpiLCs show a significant decrease in oxygen consumption and glycolysis at a 10uM concentration. Do these conditions slow cell growth? BrdU incorporation experiments (Figure 1) seem to contradict this. With compromised bioenergetics at this concentration, the authors should discuss why cell growth appears unaffected.

Indeed, we believe that cell growth is progressively restricted upon increasing doses of ∆9-THC (consider Supplementary Figure 2). In addition, oxygen consumption and glycolysis can be decoupled from cellular proliferation, especially considering the lower time ranges we are working with (44-48h).

1. Beyond Δ9-THC exposure prior to PGCLCs induction, it would be also interesting to explore the effects of Δ9-THC on PGCLCs during their differentiation.

We agree with the Reviewer. Our aim was to study whether exposure prior to differentiation could have an impact, and if so, what are the mediators of this impact. Full exposure during differentiation is another exposure paradigm that is relevant but would not have allowed us to show the metabolic memory of ∆9-THC exposure. Future work, however, will be dedicated to analyzing the effect of continuous exposure through differentiation.

1. As PGC differentiation involves global epigenetic changes, it would be interesting to investigate how Δ9-THC treatment at the ESCs/EpiLCs stage may influence PGCLCs' transcriptomes.

We also agree with the Reviewer. While this paper was not primarily focused on Δ9-THC’s epigenetic effects, we have explored the impact of Δ9-THC on more than 100 epigenetic modifiers in our RNA-seq datasets. These results are shown in Supplementary Table 1 and Supplementary Figure 10 and discussed in lines 301-316.

1. Lines 407-408: The authors should exercise caution when suggesting "potentially adverse consequences" based solely on moderate changes in PGCLCs transcriptomes.

We agree and have modified the sentence as follows: “Our results thus show that exposure to Δ9-THC prior to specification affects embryonic germ cells’ transcriptome and metabolome. This in turn could have adverse consequences on cell-cell adhesion with an impact on PGC normal development in vivo.“

1. Investigating the possible impacts of Δ9-THC exposure on cultured mouse blastocysts, implantation, post-implantation development, and fertility could yield intriguing findings.

We thank the Reviewer for this comment. We have amended our discussion to include these points in the last paragraph.

1. Given that naïve human PSCs and human PGCLCs differentiation protocols have been established, the authors should consider carrying out parallel experiments in human models.

We have performed Δ9-THC exposures in hESCs (Supplementary Figure 4 and Supplementary Figure 5), showing that Δ9-THC alters the cell number and general metabolism of these cells. We present these results in light of the differences in metabolism between mouse and human embryonic stem cells on lines 135-141 and 185-188. Implications of these results are discussed in lines 474-486.

**Reviewer #3 (Public Review):**
Verdikt et al. focused on the influence of Δ9-THC, the most abundant phytocannabinoid, on early embryonic processes. The authors chose an in vitro differentiation system as a model and compared the proliferation rate, metabolic status, and transcriptional level in ESCs, exposure to Δ9-THC. They also evaluated the change of metabolism and transcriptome in PGCLCs derived from Δ9-THC-exposed cells. All the methods in this paper do not involve the differentiation of ESCs to lineage-specific cells. So the results cannot demonstrate the impact of Δ9-THC on preimplantation developmental stages. In brief, the authors want to explore the impact of Δ9-THC on preimplantation developmental stages, but they only detected the change in ESCs and PGCLCs derived from ESCs, exposure to Δ9-THC, which showed the molecular characterization of the impact of Δ9-THC exposure on ESCs and PGCLCs.
**Reviewer #3 (Recommendations For The Authors):**
1. To demonstrate the impact of Δ9-THC on preimplantation developmental stages, ESCs are an appropriate system. They have the ability to differentiate three lineage-specific cells. The authors should perform differentiation experiments under Δ9-THC-exposure, and detect the influence of Δ9-THC on the differentiation capacity of ESCs, more than just differentiate to PGCLCs.

We apologize for the lack of clarity in our introduction. We specifically looked at the developmental trajectory of PGCs because of the sensitivity of these cells to environmental insults and their potential contribution to transgenerational inheritance. We have expanded on these points in our introduction and discussion sections (lines 89-91 and 474-486). Because our data shows the relevance of Δ9-THC-mediated metabolic rewiring in ESCs subsisting across differentiation, we agree that differentiation towards other systems (neuroprogenitors, for instance) would yield interesting data, albeit beyond the scope of the present study.

1. Epigenetics are important to mammalian development. The authors only detect the change after Δ9-THC-exposure on the transcriptome level. How about methylation landscape changes in the Δ9-THC-exposure ESCs?

We have explored the impact of Δ9-THC on more than 100 epigenetic modifiers in our RNA-seq datasets. These results are shown in Supplementary Table 1 and Supplementary Figure 10, discussed in lines 301-316. While indeed the changes in DNA methylation profiles appear relevant in the context of Δ9-THC exposure (because of *Tet2* increased expression in EpiLCs), we highlight that other epigenetic marks (histone acetylation, methylation or ubiquitination) might be relevant for future studies.

1. In the abstract, the authors claimed that "the results represent the first in-depth molecular characterization of the impact of Δ9-THC exposure on preimplantation developmental stages." But they do not show whether the Δ9-THC affects the fetus through the maternal-fetal interface.

We have addressed the need for increased clarity and have modified the sentence as follows: “These results represent the first in-depth molecular characterization of the impact of Δ9-THC exposure on early stages of the germline development.”

1. To explore the impact of cannabis on pregnant women, the human ESCs may be a more proper system, due to the different pluripotency between human ESCs and mouse ESCs.

We have performed Δ9-THC exposures in hESCs (Supplementary Figure 4 and Supplementary Figure 5). These preliminary results show that Δ9-THC exposure negatively impacts the cell number and general metabolism of hESCs. With the existence of differentiation systems for hPGCLCs, future studies will need to assess whether Δ9-THC-mediated metabolic remodelling is also carried through differentiation in human systems. We discuss these points in the last paragraph of our discussion section.

1. All the experiments are performed in vitro, and the authors should validate their results in vivo, at least a Δ9-THC-exposure pregnant mouse model.

Our work is the first of its kind to show that exposure to a drug of abuse can alter the normal development of the embryonic germline. We agree with the Reviewer that to demonstrate transgenerational inheritance of the effects reported here, future experiments in an in vivo mouse model should be conducted. The metabolic remodeling observed upon cannabis exposure could also be directly studied in a human context, although these experiments would be beyond the scope of the present study. For instance, changes in glycolysis may be detected in pregnant women using cannabis, or directly measured in follicular fluid in a similar manner as done by Fuchs-Weizman and colleagues (Fuchs-Weizman et al., 2021). We hope that our work can provide the foundation to inform such in vivo studies.